# KramaBench: A Benchmark for AI Systems on Data-to-Insight Pipelines over Data Lakes

**Eugenie Lai[1]\*, Gerardo Vitagliano[1]\*, Ziyu Zhang[1]\*, Om Chabra[1], Sivaprasad Sudhir[1],**

**Anna Zeng[1], Anton A. Zabreyko[1], Chenning Li[1], Ferdi Kossmann[1], Jialin Ding[2], Jun Chen[2]**

**Markos Markakis[1], Matthew Russo[1], Weiyang Wang[1], Ziniu Wu[1]**

**Michael J. Cafarella[1], Lei Cao[3], Samuel Madden[1], Tim Kraska[1]**
[1]MIT CSAIL, [2]Independent, [3]University of Arizona
\* denotes equal contribution. Corresponding email: `gerarvit@mit.edu`

## Abstract

Discovering insights from a real-world data lake potentially containing unclean, semi-structured, and unstructured data requires a variety of data processing tasks, ranging from extraction and cleaning to integration, analysis, and modeling. This process often also demands domain knowledge and project-specific insight. While AI models have shown remarkable results in reasoning and code generation, their abilities to design and execute complex pipelines that solve these data-lake-to-insight challenges remain unclear. We introduce **KramaBench**[1] which consists of 104 manually curated and solved challenges spanning 1700 files, 24 data sources, and 6 domains. **KramaBench** focuses on testing the end-to-end capabilities of AI systems to solve challenges which require automated orchestration of different data tasks. **KramaBench** also features a comprehensive evaluation framework assessing the *pipeline design* and *individual data task implementation* abilities of AI systems. We evaluate 8 LLMs using our single-agent reference framework DS-Guru, alongside both open- and closed-source single- and multi-agent systems, and find that while current agentic systems may handle isolated data-science tasks and generate plausible draft pipelines, they struggle with producing working end-to-end pipelines. On **KramaBench**, the best system reaches only 55% end-to-end accuracy in the full data-lake setting. Even with perfect retrieval, the accuracy tops out at 62%. Leading LLMs can identify up to 42% of important data tasks but can only fully implement 20% of individual data tasks.

## 1 Introduction

The goal of data science is to obtain insights from raw data. A data science workflow typically involves manually selecting data and designing pipelines that perform data wrangling, conduct data analyses, and extract findings, among other data tasks. These workflows (Figure 1) are expected to handle noisy, domain-specific data and scale to data lakes with tens to thousands of files, necessitating multi-step, data-dependent reasoning and coordination across data tasks (Guo et al., 2024; Shankar et al., 2025).

While recent research has advanced individual components of these workflows such as code generation (Nam et al., 2024; Wang & Chen, 2023), tool use (Qin et al., 2024b;a), and natural language question answering (Zhang et al., 2024b; Pu et al., 2023), the challenge of designing and executing complete end-to-end data science pipelines remains underexplored.

Progress towards practical data-to-insight systems has been hindered by the lack of benchmarks that reflect the real-world complexity of these workflows. Existing benchmarks focus on isolated steps,

---

[1]Assets available at `https://github.com/mitdbg/Kramabench`

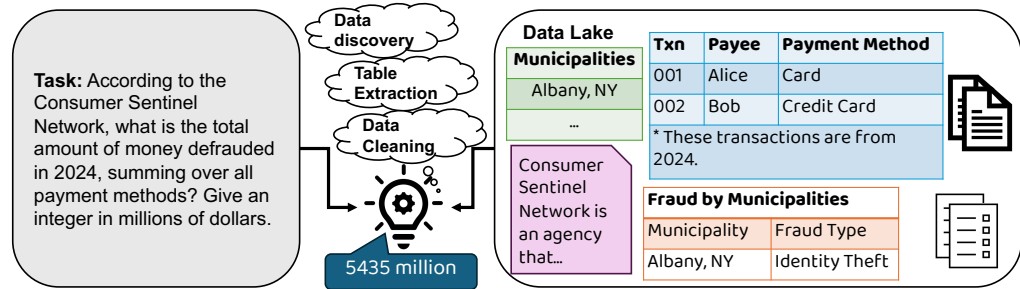

Figure 1: One of the tasks of **KRAMABENCH** based on a real data lake of 136 files in the legal discovery domain. Data file sample snippets are simplified.

Table 1: Comparing existing benchmarks. (– indicates partial satisfaction, e.g., not for all tasks)

| Benchmarks | DS-1000 Lai et al. (2023) | ARCADE Yin et al. (2023) | DA-Code Huang et al. (2024) | DataSciBench Zhang et al. (2025a) | DSBench Jing et al. (2025) | BLADE Gu et al. (2024) | ScienceAgentBench Chen et al. (2025) | Ours |
|---|---|---|---|---|---|---|---|---|
| **DS Tasks** | | | | | | | | |
| Data discovery | ✗ | ✗ | ✗ | ✗ | ✗ | ✗ | ✗ | ✔ |
| Multi-file integration | ✗ | ✔ | ✔ | ✗ | ✔ | ✗ | ✗ | ✔ |
| Data cleaning | ✔ | ✔ | ✔ | ✔ | ✗ | ✗ | ✗ | ✔ |
| Data preparation | ✔ | ✔ | ✔ | ✔ | ✔ | ✔ | ✔ | ✔ |
| Data analysis | ✔ | ✔ | ✔ | ✔ | ✔ | ✔ | ✔ | ✔ |
| Modeling | ✔ | ✔ | ✔ | ✔ | ✔ | ✔ | ✔ | ✔ |
| **Abilities tested** | | | | | | | | |
| Data semantics | ✗ | ✗ | ✗ | ✗ | – | – | ✗ | ✔ |
| Domain knowledge | ✗ | ✗ | ✗ | ✗ | ✗ | ✔ | ✔ | ✔ |
| Multi-step reasoning | ✔ | ✔ | ✔ | ✔ | ✔ | ✔ | ✔ | ✔ |
| **Evaluation** | | | | | | | | |
| Implementation | ✔ | ✔ | ✔ | ✔ | ✔ | ✔ | ✔ | ✔ |
| Pipeline design | ✗ | ✗ | ✗ | ✗ | – | – | ✗ | ✔ |
| End-to-end | ✗ | ✗ | ✗ | ✗ | ✗ | ✗ | ✗ | ✔ |

such as code generation from fine-grained prompts (Lai et al., 2023; Zhang et al., 2025a; Huang et al., 2024; Yin et al., 2023), text-to-SQL (Lei et al., 2025; Zhang et al., 2024a), and modeling using curated input (Gu et al., 2024; Mitchener et al., 2025; Chen et al., 2025). We list these works in Table 1 and discuss more in Section 5. While immensely useful, these benchmarks do not capture the heterogeneity of data tasks and the accompanying reasoning demands of real-world data science involving large, domain-specific, and unclean input datasets.

To bridge this gap, we introduce **KRAMABENCH**[2], a benchmark designed to evaluate LLM-based systems on complex end-to-end data science pipelines. **KRAMABENCH** consists of 104 tasks drawn from 1700 real-world files across 24 sources in 6 domains. All tasks are manually curated from fresh, domain-specific sources and paired with expert reference solutions grounded in accessible data. Each task is specified in natural language and requires systems to discover relevant data, perform data wrangling such as cleaning and normalization, and implement statistical or computational analyses to produce insights. To study public data's leakage into LLM training, we obscured the input of 20% of tasks through replacing real-world identifiers and numeric data with synthetic ones without changing the task structure. We hold them out for evaluation to prevent them from being trained on.

For each task, we provide reference sub-tasks that a system capable of solving the end-to-end task should be able to solve. Sub-tasks are also annotated with ground truth results and text descriptors. These assets facilitate our comprehensive evaluation framework with three settings. (1) The most important *end-to-end automation* setting assesses the ability to solve tasks without a human in the loop. (2) The *pipeline design* setting assesses the ability to reason and identify key components towards a successful pipeline design. (3) The *individual task implementation* setting assesses the ability to act on fine-grained descriptions of individual sub-tasks in a correct pipeline.

---

[2]The name KramaBench is a reference to the "Vinyasa Krama" practice of Yoga

We evaluated **KRAMABENCH** across eight models, along with three different configurations of DS-Guru and three other existing agentic systems (Hugging Face, 2025; OpenAI, 2025; Google, 2025). We conducted extensive ablations studies and failure analyses, taking advantage of our comprehensive evaluation framework and obscured inputs.

Through **KRAMABENCH**, we observed multiple insights about where LLM systems are successful: (1) Agentic control flow is helpful with **KRAMABENCH**'s challenges: a canonical single-agent system (*smolagents DR*) that iteratively search, plan, and repair achieve $55.83\%$ end-to-end accuracy, outperforming the strongest configurations of DS-Guru ($24.98\%$ overall), which uses a structured control flow. (2) a canonical multi-agent system (*smolagents Reflexion*, Shinn et al. (2023)) using an evaluator agent and reflections achieves $55.37\%$ end-to-end accuracy. The marginal difference ($-0.45\%$) relative to the single-agent baseline suggests that simple evaluator-style coordination alone does not substantially improve performance on data-intensive workflows, indicating opportunities for more specialized multi-agent designs.(3) LLM systems can reason at a coarse level about the data operations required by a successful workflow and generate plausible pipelines, achieving $41.71\%$ on pipeline design.

Our analyses also reveal some persistent challenges: (1) Retrieval from a data lake is problematic, but not the dominating obstacle. Supplying only the gold files improves overall accuracy by only $0 - 7\%$ across systems using different retrieval mechanisms. (2) Weaknesses in fine-grained data-dependent reasoning cause models to fail. Systems even fail most of the time at implementing individual simple sub-tasks, capping at $22.05\%$ when evaluated under the individual task implementation described above. (3) Agents often fail to achieve a holistic understanding of the data lake. We observe that the agents often overly rely on their prior knowledge (at most $43.06\%$ performance fluctuation on obscured inputs), or assume clarifications will be given from a user ($24\%$ of failures).

## 2 THE DESIGN OF **KRAMABENCH**

Tasks in **KRAMABENCH** are based on real-world data science challenges from six domains: archeology, astronomy, biomedical research, environmental science, legal insight discovery, and wildfire prevention. Each domain is associated with a data lake containing raw files in structured, semi-structured, or unstructured formats from multiple sources. Each **task** is a natural language description of a domain-specific data science problem. The goal of a system under test is to design and execute an end-to-end pipeline that takes the entire domain data lake as input and produces the correct output. In addition to the target answer, **KRAMABENCH** provides the ground truth solution both in code and in annotated **sub-tasks**: natural language descriptions of smaller building-block operations that are essential elements within a full solution along with a prompt and their target answers. These finer-grained references enable the evaluations of pipeline design and individual task implementation.

### 2.1 TASK DESIGN AND VALIDATION

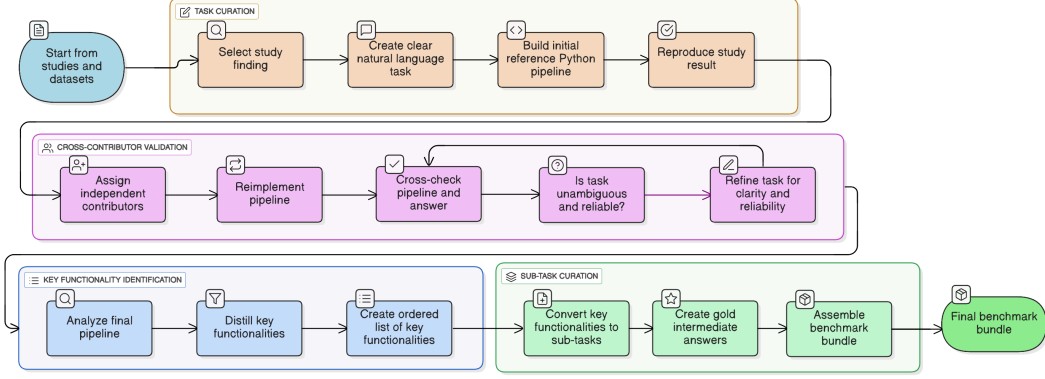

Figure 2: Workflow for task design and validation in **KRAMABENCH**, detailing the curation, validation, and functional decomposition to ensure quality and consistency across tasks.

Table 2: Detailed breakdown of per-domain tasks in **KRAMABENCH**. Hard tasks require multiple files or pipelines with more than three steps.

| Domain | # Tasks (sub-) | %Hard Tasks | # Files (size) |
|---|---|---|---|
| Archeology | 12 (71) | 50.00% | 5 (7.5MB) |
| Astronomy | 12 (68) | 50.00% | 1556 (486MB) |
| Biomedical | 9 (38) | 66.66% | 7 (175MB) |
| Environment | 20 (148) | 70.00% | 37 (31MB) |
| Legal | 30 (188) | 53.33% | 136 (1.3MB) |
| Wildfire | 21 (120) | 71.42% | 23 (1GB) |
| **Total** | 104 (633) | 60.58% | 1764 (1.7GB) |

Table 3: Answer type and example questions.

| Type | Metric score |
|---|---|
| String (exact) | Accuracy (0/1) |
| String (approximate) | ParaPluie score (0/1) |
| Numeric (exact) | Accuracy (0/1) |
| Numeric (approximate) | $1/(1 + \text{RAE})$ (0-1) |
| List (exact) | F1 score (0-1) |
| List (approximate) | F1 score (if match > 0.9) |

To curate tasks, we started with published studies and reports that (1) contain quantitative or graphical findings produced by data analysis, (2) are based on complete and publicly accessible datasets, and (3) require complex multi-step pipelines involving heterogeneous and noisy inputs. Grounding onto these studies and reports ensures that our tasks reflect real-world data science pipelines. We followed a 4-step workflow involving tight validations and repeated verifications of reference solutions to ensure the quality of tasks, reference solutions, and fine-grained annotations. We summarize the process in Figure 2.

**Step 1: Task Curation.** For each study or report, we reproduced its important findings using the associated datasets, transforming these findings into problem statements. Within the same domain, more tasks similar to the real-world ones are curated via integrating different data sources. The creator of each task supplies a concrete implementation of the pipeline.

**Step 2: Cross-Contributor Validation.** For each task, a different second contributor independently attempts to develop a solution. A third contributor compares the solution with the one in Step 1., resolves ambiguities in the problem statement, and checks in a reference pipeline. The execution time of the reference pipeline is also recorded.

**Step 3: Key Functionality Identification.** A data science problem can have multiple valid solution pipelines. However, certain data processing steps *must* exist in any correct pipeline. A simple example would be "*identifying the column containing the temperature to* `Temp`". We draft a list of these key functionalities for each task using the reference pipeline via instruction-tuning GPT-o3 and manually polish the outputs to make sure the description of these sub-tasks do not depend on specific implementation choices. The semi-automation scripts are available at our repository.

**Step 4: Sub-task Curation.** We transform each sub-task description into a prompt via instruction tuning a local instance of Gemma3-27b and manual inspection. The example in Step 3 would be transformed to "*which column contains the temperature information*"? The target answers to each sub-task are manually verified using the reference pipeline.

Table 2 reports the statistics and difficulty distributions of the 6 domains and their tasks. We provide more detailed descriptions and an example of tasks, key functionalities, and sub-tasks in Appendix E.

## 2.2 EVALUATION MECHANISM

As discussed in Section 1, **KRAMABENCH** evaluates systems on three capabilities. In Figure 3, we provide an overview of these 3 areas. Our primary focus is (1) end-to-end automation.

**(1) End-to-end Automation.** For each task, the system output is given a **score** in $[0, 1]$ based on the reference target answer. The scoring schemes for each possible answer type address fuzzy matches and are discussed in Table 3. The string approximation metric uses the method introduced in (Lemesle et al., 2025). Our validation study (details in Subsection G.2) for this LLM-as-a-judge method shows 84% agreement between human annotators and the LLM. Given a domain workload $W$ consisting of numerous tasks $T's$, the **total** score of a system $\mathcal{F}$ for $W$ is $\text{Mean}_{T \in W} \text{score}(\mathcal{F}(T))$. The score of $\mathcal{F}$ for the entire benchmark suite is analogous.

Results under the following two less-automated evaluation settings provide insights into why a system may succeed or fail in the end-to-end automation setting and the abilities of a system to assist with a human-in-the-loop. Figure 8 describes the mapping of the following tasks.

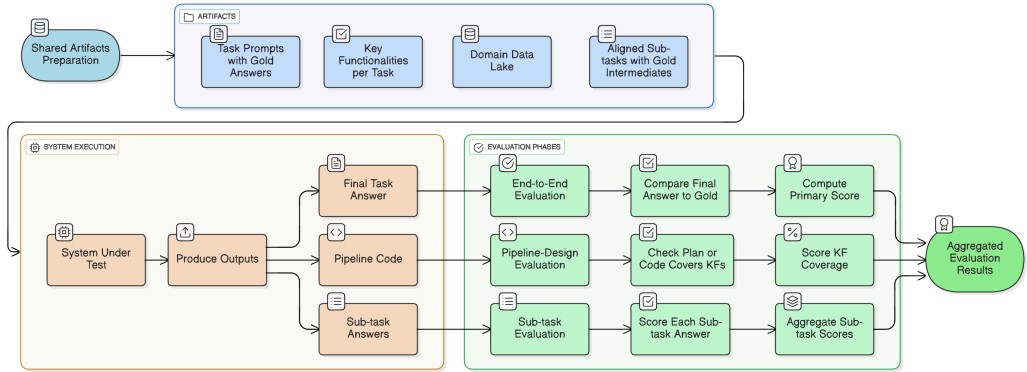

Figure 3: Overview of **KRAMABENCH**'s evaluation process.

**(2) Pipeline Design.** For each task, we assess the system generated pipeline using the key functionalities that *any correct pipeline* needs to contain in some form (from Step 3 of Subsection 2.1). We score the system with the fraction of key functionalities covered in the pipeline produced by the system. Coverage is evaluated via LLM-as-a-judge following the method in Tong & Zhang (2024) using the description obtained in Step 3.

**(3) Sub-task Evaluation.** We provide systems with the problem statements for sub-tasks and compare system outputs to human-curated target answers (as in Step 4 of Subsection 2.1) using the same scoring approach as in end-to-end evaluation. Full technical details of these evaluations are provided in Appendix G.

## 2.3 REFERENCE IMPLEMENTATION

We introduce DS-Guru (Figure 4), a lightweight framework that serves as minimal scaffolding to enable a single out-of-the-box LLM to attempt the data science challenges in **KRAMABENCH**. DS-Guru has three variants. *No-context:* The LLM is invoked one-shot with the problem description and the names and paths of the files from the data lake, without any file contents. *One-shot:* The LLM is invoked one-shot with the problem description and sample snippets from each data file. *Few-shot:* The LLM is first invoked once with the task description and sample snippets, then re-invoked few-shot with execution results and error messages from the pipeline it implemented in the previous shot. With all variants, DS-Guru instructs the LLM to decompose the task into simpler tasks before attempting to implement each task provide the concrete pipeline implementation along with the answer.

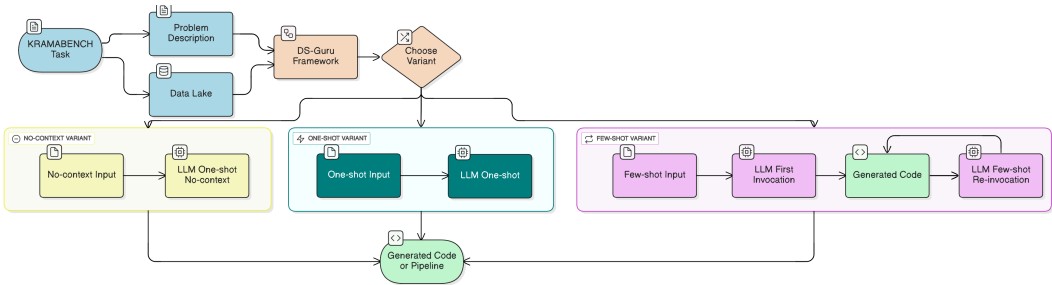

Figure 4: DS-Guru, a lightweight framework that scaffolds out-of-the-box LLMs with multiple variants to tackle **KRAMABENCH**.

DS-Guru succinctly addresses where out-of-the-box LLMs struggle with **KRAMABENCH**. (1) Realistic data lakes exceed LLM context windows. DS-Guru uses budgeted, type-annotated **one-pass sampling (OPS)** retrieval to make this step tractable. (2) Many data science tasks require many different data operations. DS-Guru uses chain-of-thought prompting (Wei et al., 2022) to encourage decomposition before code synthesis. (3) Code running on real-world uncurated data are subject to more sporadic errors compared to code for well-structured tasks. DS-Guru's multi-shot approach (Press et al., 2023) can help LLMs recover from such errors. More details on DS-Guru in Appendix B.

## 3 EXPERIMENTAL SETUP

We accessed all LLMs in different systems via OpenAI and Together APIs; pipelines generated by systems are executed locally. Whenever possible, we repeated all experiments three times and report average score and standard deviations across the three runs. For some results, marked with an asterisk, we could not perform repeated experiments - due to high API costs or to the backend models being discontinued during the experiments.

**DS-Guru:** We combine each of the three variants (as in Subsection 2.3) of DS-Guru with six LLMs: GPT-o3, GPT-4o, Claude-3.5-Sonnet, Llama3.3, Deepseek-R1-70B, and Qwen2.5-Coder-32B (OpenAI, 2025; 2024; Anthropic, 2024; Meta AI, 2025; DeepSeek-AI et al., 2025; Hui et al., 2024), totaling to 18 concrete DS-Guru implementations.

**smolagents Deep Research (smolagents DR):** We use Hugging Face `smolagents` (Hugging Face, 2025) to evaluate deep research-style agentic systems on our benchmark under both single-agent and multi-agent settings. On the single agent setting (smolagents-single DR), `smolagent's` official single-agent deep research implementation using code actions (git), we report results with two different LLMs as the LLM: GPT-o3 and Claude-3.7-Sonnet. For the multi-agent setting, we implemented two representative architectures also using `smolagents`: (1) smolagents Reflexion (Shinn et al., 2023) which follows an actor → evaluator → reflection agent loop. (2) smolagents PDT which follows a **P**lanner and Task **D**ecomposer → **T**ool Executor workflow (PDT), inspired by Fan et al. (2025). We provide more details in Appendix C. We view these systems representative of open-source "deep research" projects (e.g., Alibaba's Academy (2025)).

**Closed-Source Deep Research Systems: OpenAI Deep Research (OpenAI DR**, OpenAI (2025)) and **Gemini Pro-2.5 Agentic Mode (Gemini Agentic)** (Google DeepMind, 2025; Google, 2025) were evaluated manually through their web interfaces under the end-to-end automation setting. We made best efforts instructing them not to search online. However, this restriction was not enforceable.

Table 4: Comparison of different mechanisms across systems.

| Systems | Retrieval mechanisms | Input modes | Control flow | Internet Access |
|---|---|---|---|---|
| DS-Guru | One-Pass Sampling (OPS) | Full, Trimmed, Oracle | Structured loops | Off |
| smolagents DR | Agentic retrieval | Full, Trimmed, Oracle | Single-agent ReAct loop | Off |
| smolagents Reflexion | Agentic retrieval | Full, Trimmed, Oracle | Multi-agent | Off |
| smolagents PDT | Agentic retrieval | Full, Trimmed, Oracle | Multi-agent | Off |
| OpenAI DR | Agentic retrieval | Trimmed, Oracle* | Agentic loops | On |
| Gemini Agentic | Agentic retrieval | Trimmed, Oracle* | Agentic loops | On |

**Human Baseline.** We conducted a small-scale human study in which 9 data-science practitioners solved **KRAMABENCH** under the same conditions and requirements as LLM systems under test in the end-to-end, full input setting. Details can be found in Appendix F.

We evaluated six different systems, which differ in four important ways.

**Retrieval Mechanisms.** DS-Guru employs *One-Pass Sampling* (*OPS*) retrieval: a budgeted, type-annotated sample of each file in the data lake (schema summaries + a small row sample) is provided to the LLM once. *OPS* scales with data lake size but constrains the LLM's direct interaction to sampled views. DR systems employ *agentic retrieval*: the LLM plans the retrieval and issues file system tool calls to iteratively read, filter, and revisit sources, offering richer interaction but at a higher cost.

**Input Modes.** *Full*: ideally, the entire input lake is available to the system's retriever. *Oracle*: only the gold files are provided (no discovery), isolating non-retrieval failures (planning, reasoning, execution). *Trimmed*: to respect practical constraints, most notably the UI limit of $\leq 10$ file uploads imposed by the closed-source DR systems, we supply the gold files plus a random subset of distractors up to the limit, testing discovery under budget. *Oracle*: for tasks where the gold set itself exceeds 10 files, we include the task by randomly sampling 10 gold files for upload.

Control flow describes whether a system has fully structured loops or an agentic workflow where agents decide the future courses of actions. Internet access describes whether systems have web search capabilities.

**Cost of evaluation.** For DS-Guru (few-shot, GPT-o3), evaluating end-to-end answers, pipeline design, and sub-task implementation took 4,501, 116,805, and 10,358 tokens respectively.

# 4 RESULTS AND TAKEAWAYS

Table 5 shows the performance of the systems under the *Full*, *Oracle*, and *Trimmed* input mode. We report only top-performing configurations here and present full results in Appendix A.

Table 5: Results by domain for **KRAMABENCH** on DS-Guru and smolagents for three input settings. Results marked with an asterisk are not averaged across three executions (mean ± standard deviation).

| System | Models | Domains | | | | | | |
|---|---|---|---|---|---|---|---|---|
| | | Archeology | Astronomy | Biomedical | Environment | Legal | Wildfire | Overall |
| **Full Input Mode** | | | | | | | | |
| Human baseline* | | 58.33 | 70.00 | 100.00 | 76.90 | 86.67 | 66.20 | 76.75 |
| DS-Guru no-context | GPT-o3 | 16.67 ± 0.00 | 0.00 ± 0.00 | 0.10 ± 0.14 | 5.00 ± 4.08 | 0.00 ± 0.00 | 14.18 ± 4.28 | 5.87 ± 0.71 |
| | GPT-4o | 0.00 ± 0.00 | 0.00 ± 0.00 | 0.00 ± 0.00 | 0.00 ± 0.00 | 0.00 ± 0.00 | 0.00 ± 0.00 | 0.00 ± 0.00 |
| | Claude-3.7 | 2.78 ± 3.93 | 3.33 ± 4.71 | 0.00 ± 0.00 | 0.00 ± 0.00 | 1.11 ± 1.57 | 4.38 ± 3.92 | 1.88 ± 0.81 |
| DS-Guru one-shot | GPT-o3 | 19.44 ± 3.93 | 0.00 ± 0.00 | 0.53 ± 0.75 | 16.11 ± 9.26 | 10.00 ± 0.00 | 40.01 ± 4.22 | 16.67 ± 2.93 |
| | GPT-4o | 8.33 ± 6.80 | 6.67 ± 4.71 | 3.70 ± 0.00 | 0.00 ± 0.00 | 4.41 ± 2.17 | 13.69 ± 3.06 | 6.08 ± 1.25 |
| | Claude-3.7 | 2.78 ± 3.93 | 0.00 ± 0.00 | 0.00 ± 0.00 | 3.33 ± 2.36 | 0.00 ± 0.00 | 1.59 ± 2.24 | 1.31 ± 1.22 |
| DS-Guru few-shot | GPT-o3 | 13.89 ± 3.93 | 0.00 ± 0.00 | 0.10 ± 0.14 | 49.56 ± 0.87 | 9.26 ± 2.92 | 52.92 ± 0.72 | 24.98 ± 1.25 |
| | GPT-4o | 13.89 ± 3.93 | 3.33 ± 4.71 | 0.00 ± 0.00 | 0.00 ± 0.00 | 4.84 ± 3.16 | 25.69 ± 1.15 | 8.67 ± 1.48 |
| | Claude-3.7 | 5.56 ± 3.93 | 0.00 ± 0.00 | 0.00 ± 0.00 | 15.00 ± 4.08 | 8.89 ± 6.29 | 10.09 ± 9.67 | 8.29 ± 1.12 |
| smolagents DR | GPT-o3 | 33.33 ± 11.79 | 23.33 ± 9.43 | 37.04 ± 10.48 | 31.33 ± 8.96 | 17.78 ± 8.31 | 35.23 ± 11.46 | 28.07 ± 8.80 |
| | Claude-3.7 | **44.44 ± 3.93** | **50.00 ± 8.16** | **38.89 ± 7.86** | **60.56 ± 7.74** | **61.23 ± 12.70** | **60.16 ± 7.43** | **55.83 ± 3.41** |
| smolagents Reflexion | GPT-o3 | 30.56 ± 10.39 | 46.67 ± 4.71 | 29.63 ± 5.24 | 23.89 ± 14.55 | 26.67 ± 14.40 | 35.04 ± 8.36 | 30.53 ± 10.79 |
| | Claude-3.7 | 41.67 ± 0.00 | 45.00 ± 5.00 | 50.00 ± 27.78 | 56.25 ± 8.75 | 58.33 ± 1.67 | 65.38 ± 0.86 | 55.37 ± 3.36 |
| smolagents PDT | GPT-o3 | 8.33 ± 6.80 | 0.00 ± 0.00 | 0.00 ± 0.00 | 1.42 ± 1.09 | 4.51 ± 0.84 | 22.01 ± 1.39 | 7.57 ± 1.28 |
| | Claude-3.7 | 19.44 ± 7.86 | 6.67 ± 4.71 | 5.56 ± 5.56 | 9.84 ± 2.86 | 6.25 ± 4.95 | 22.58 ± 4.05 | 12.01 ± 1.03 |
| **Oracle Input Mode** | | | | | | | | |
| DS-Guru no-context | GPT-o3 | 11.11 ± 3.93 | 10.00 ± 8.16 | 11.11 ± 0.00 | 0.00 ± 0.00 | 0.00 ± 0.00 | 10.17 ± 0.92 | 5.36 ± 1.20 |
| | GPT-4o | 0.00 ± 0.00 | 0.00 ± 0.00 | 0.00 ± 0.00 | 0.00 ± 0.00 | 0.00 ± 0.00 | 0.00 ± 0.00 | 0.00 ± 0.00 |
| | Claude-3.7 | 4.17 ± 4.17 | 10.00 ± 0.00 | 0.00 ± 0.00 | 1.58 ± 1.58 | 1.67 ± 1.67 | 8.96 ± 0.57 | 4.11 ± 0.79 |
| DS-Guru one-shot | GPT-o3 | 8.33 ± 6.80 | 16.67 ± 4.71 | 7.51 ± 5.31 | 23.89 ± 6.71 | 15.56 ± 1.57 | 42.80 ± 3.48 | 21.35 ± 0.84 |
| | GPT-4o | 11.11 ± 7.86 | 13.33 ± 4.71 | 5.19 ± 7.33 | 12.33 ± 3.30 | 6.67 ± 0.00 | 19.85 ± 3.61 | 11.54 ± 1.47 |
| | Claude-3.7 | 0.00 ± 0.00 | 0.00 ± 0.00 | 0.00 ± 0.00 | 6.67 ± 1.67 | 3.33 ± 3.33 | 4.76 ± 0.00 | 3.27 ± 1.31 |
| DS-Guru few-shot | GPT-o3 | 16.67 ± 0.00 | 26.67 ± 4.71 | 7.41 ± 5.24 | 50.11 ± 4.09 | 61.03 ± 5.78 | 52.02 ± 3.79 | 43.71 ± 1.94 |
| | GPT-4o | 13.89 ± 3.93 | 20.00 ± 0.00 | 3.70 ± 5.24 | 18.44 ± 6.00 | 37.78 ± 4.16 | 36.67 ± 3.53 | 26.20 ± 2.34 |
| | Claude-3.7 | 20.83 ± 4.17 | 25.00 ± 15.00 | 0.00 ± 0.00 | 9.17 ± 9.17 | 31.67 ± 1.67 | 18.17 ± 5.20 | 19.75 ± 3.36 |
| smolagents DR | GPT-o3 | 27.78 ± 7.86 | 26.67 ± 4.71 | 37.04 ± 5.24 | 24.00 ± 0.00 | 17.78 ± 5.67 | 32.89 ± 10.31 | 25.86 ± 4.57 |
| | Claude-3.7 | **45.83 ± 12.50** | **65.00 ± 5.00** | 58.33 ± 2.78 | 46.83 ± 1.50 | 65.00 ± 1.67 | **75.08 ± 3.78** | 60.67 ± 1.23 |
| smolagents Reflexion | GPT-o3 | 30.56 ± 3.93 | 40.00 ± 0.00 | 44.44 ± 0.00 | 31.44 ± 2.20 | 22.22 ± 8.31 | 39.78 ± 6.14 | 32.33 ± 4.44 |
| | Claude-3.7 | 37.50 ± 4.17 | 50.00 ± 0.00 | **83.33 ± 5.56** | **62.67 ± 4.33** | **70.00 ± 10.00** | 64.45 ± 1.39 | **62.81 ± 3.10** |
| smolagents PDT | GPT-o3 | 8.33 ± 6.80 | 0.00 ± 0.00 | 0.00 ± 0.00 | 3.26 ± 1.30 | 8.62 ± 3.88 | 15.86 ± 5.53 | 7.69 ± 2.69 |
| | Claude-3.7 | 16.67 ± 0.00 | 10.00 ± 0.00 | 11.11 ± 0.00 | 11.50 ± 3.50 | 9.24 ± 0.35 | 25.50 ± 5.46 | 14.38 ± 2.15 |
| **Trimmed Input Mode** | | | | | | | | |
| DS-GURU few-shot | GPT-o3 | 25.00 ± 6.80 | 0.00 ± 0.00 | 3.70 ± 5.24 | 48.33 ± 8.50 | 51.11 ± 3.14 | 48.87 ± 7.90 | 37.84 ± 3.90 |
| smolagents-single DR | Claude-3.7* | **50.00** | **80.00** | **55.56** | 40.83 | **60.00** | 67.24 | **58.12** |
| OpenAI Online Deep Research* | | 40.00 | 33.33 | 44.45 | **61.67** | 50.00 | **67.28** | 52.18 |
| Google Agentic Gemini-2.5-Pro* | | 25.00 | 16.67 | 33.33 | 25.00 | 13.33 | 24.87 | 18.48 |

**Agentic control flows drive the largest performance gains on KRAMABENCH.** The smolagents DR variant using Claude-3-7 consistently outperforms DS-Guru across all domains (Table 5), reaching an average $55.83\%$ overall score compared to the best DS-Guru variant (few-shot, GPT-o3; $24.98\%$). The DS-Guru few-shot variant, which enables the LLM to catch implementation errors, improves over the one shot variant, e.g., using GPT-o3, with a $8.31\%$ overall improvement. Our detailed studies increased few-shot to 20 iterations yet still showed minor improvements (Appendix Table 13). This indicates that despite the heterogeneity of data operations, the core challenges are not isolated data operation implementation issues, but instead are to (1) explore and fix the design choices of the end-to-end pipeline; (2) iteratively understand the data and schema in a large data lake. Smolagent DR's agentic control flow helps address these challenges. In the *Trimmed* setting (max 10 files per call), OpenAI's own online DeepResearch tool reaches $58.12\%$ overall, partly due to its web search capability. We refer the reader to Appendix F for detailed analysis of the human baseline., as well as for a detailed analysis of cost and runtimes of each models. In terms of cost, smolagents DR (Claude-3-7) averages 6.10 minutes per task—faster than OpenAI DR (10.35) but more than 10× slower than DS-Guru few-shot (0.76).

## 4.1 ABLATION STUDIES

**Retrieval Mechanisms.** Overall, using *Oracle* input for DS-Guru improves the performance across all domains and LLMs (by $6.38\%$ on average and up to $18.73\%$) except for GPT-o3 for which in the

Table 6: DS-Guru (few-shot, 5 iterations, GPT-o3): performance and cost across rows sampled.

| Rows Sampled | Overall Score (%) | Tokens (Mean) |
|---|---|---|
| 10 | 22.89 | 14,077.2 |
| 50 | 24.68 | 37,592.3 |
| 100 | 23.36 | 64,548.9 |
| 150 | 22.58 | 92,116.1 |

Table 7: End-to-end scores of various systems under obscured vs oracle inputs (mean ± standard deviation).

| System | Model | Full | Oracle | Obscured |
|---|---|---|---|---|
| DS-Guru no-context | GPT-o3 | 5.87±0.71 | 5.36±1.20 | 3.69±1.63 |
| | Claude-3.7 | 1.88±0.81 | 4.11±0.79 | 0.00±0.00 |
| DS-Guru one-shot | GPT-o3 | 16.67±2.93 | 21.35±0.84 | 7.85±1.27 |
| | Claude-3.7 | 1.31±1.22 | 3.27±1.31 | 0.50±0.50 |
| DS-Guru few-shot | GPT-o3 | 24.98±1.25 | 43.71±1.94 | 23.02±0.49 |
| | Claude-3.7 | 8.29±1.12 | 19.75±3.36 | 3.80±3.80 |
| smolagents DR | GPT-o3 | 28.07±8.80 | 25.86±4.57 | 10.07±0.30 |
| | Claude-3.7 | 55.83±3.41 | 60.67±1.23 | 12.77±0.00 |
| smolagents Reflexion | GPT-o3 | 30.53±10.79 | 32.33±4.44 | 9.52±3.95 |
| | Claude-3.7 | 55.37 ± 3.36 | 62.81±3.10 | 13.94±0.00 |
| smolagents PDT | GPT-o3 | 7.57±1.28 | 7.69±2.69 | 3.87±3.52 |
| | Claude-3.7 | 12.01±1.03 | 14.38±2.15 | 2.23±0.00 |

no-context scenario no significant improvement is registered (Table 5). These results under the design of DS-Guru show that supplying samples of the gold files can lead to more successful pipelines. We also studied the sensitivity of *OPS* against the sample size from each file. Table 6 shows that the performance of the system does not meaningfully increase with larger samples.

The benefits of the *Oracle* in smolagents DR show the same trend, suggesting that *agentic retrieval is not qualitatively closer to perfect retrieval than OPS in terms of file extraction*. Even with the *Oracle* input, the agentic smolagents DR with out-of-the-box LLMs still struggle to solve a large portion of the tasks (62.81% overall with Claude-3.7). These results point to weaknesses in data-dependent reasoning (e.g., pipeline design), in addition to extracting the right files.

**Data Leakage.** To study to what extent different systems are solving tasks via external knowledge present in previous knowledge data instead of producing a reliable data pipeline, we manually curated **obscured inputs** for all tasks in **KRAMABENCH**, where data fields are changed such that a correct pipeline would still produce a correct solution, but a system relying on memorization cannot. For example, in a query spanning multiple locations, the real place names may be swapped for fictional ones, i.e., Los Angeles might be changed to "La-La Land."

For both smolagents single DR and DS-Guru few-shot, the performance under the obscured input is $15 - 18\%$ lower compared to the regular and oracle inputs (Table 7). Interestingly, Claude as a backend model seems the most sensitive to the obscured workloads, e.g. with a sharp drop from 62.81% to 12.77% in the Reflexion baseline, suggesting that the solution of the benchmark in these variants may heavily rely on parametric knowledge. These observations and the stark difference between the *Full* and *Obscured* input performances suggest two distinctive plausible explanations for our observations: (1) Prior knowledge could discourage attempts at data-dependent reasoning. (2) Prior knowledge could be serving as an unintended reward signal in agentic data-dependent reasoning, which possibly can either reduce the performance of the system.

**Cross-domain Accuracy Difference** The per-domain accuracy of the best performing system (smolagents Reflexion) in **KRAMABENCH** varies as much as 41.67% on Archeology and 65.38% on Wildfire. Our analysis of system traces show that the primary source of these cross-domain accuracy differences is the differences in the types of data task challenges that each domain emphasize on. We discuss this in more detail in Subsection D.1.

**Diversity of Abilities Required from LLM Agents.** Tested independently, pipeline design and subtask implementation outperformed end-to-end automation, e.g, even in the no-context scenario, using GPT-o3 as a model, the pipeline design score is 39.75% compared to 5.87% end-to-end, In addition, we observe that LLMs also have varying capability profiles: GPT-o3 is strong at high-level pipeline design (39.75%) but weak at implementing those pipelines $(10 - 22\%)$; interestingly, it scores higher on end-to-end automation (24.98%) than on some implementation tasks. Although with worse absolute results, DeepSeek-R1 exhibits the opposite pattern (1% on pipeline design vs 5.79% on implementation). These patterns provide strong evidence that single-model approaches are insufficiently reliable for real-world data science, as success depends on multiple heterogeneous

Table 8: Lower automation settings results on KRAMABENCH (mean ± standard deviation).

| | | Models | | | |
|---|---|---|---|---|---|
| Variant | Automation setting | GPT-o3 | GPT-4o | Llama3 Instruct | DeepSeek-R1 |
| DS-Guru no-context | End-to-end automation | 5.87 ± 0.71 | 0.00 ± 0.00 | 0.33 ± 0.46 | 0.33 ± 0.46 |
| | Pipeline Design | 39.75 ± 0.73 | 29.73 ± 1.16 | 21.80 ± 1.30 | 0.63 ± 0.47 |
| | Sub-task Implementation | 10.69 ± 0.35 | 3.32 ± 0.54 | 3.88 ± 0.25 | 1.48 ± 0.08 |
| DS-Guru one-shot | End-to-end automation | 16.67 ± 2.93 | 6.08 ± 1.25 | 3.42 ± 0.37 | 0.00 ± 0.00 |
| | Pipeline Design | 41.71 ± 1.00 | 21.26 ± 0.64 | 15.53 ± 0.73 | 1.59 ± 0.77 |
| | Sub-task Implementation | 17.33 ± 0.48 | 5.08 ± 0.57 | 2.80 ± 0.08 | 3.91 ± 0.44 |
| DS-Guru few-shot | End-to-end automation | 24.98 ± 1.25 | 8.67 ± 1.48 | 8.40 ± 1.01 | 0.33 ± 0.46 |
| | Pipeline Design | 37.99 ± 2.64 | 20.31 ± 0.07 | 12.81 ± 1.04 | 1.11 ± 0.72 |
| | Sub-task Implementation | 22.05 ± 0.71 | 6.60 ± 0.29 | 5.53 ± 0.49 | 5.79 ± 0.72 |

skills, such as robust parsing of noisy inputs, query parsing and planning, identifying and performing data-cleaning/ transformation, coding, and iterative debugging.

## 4.2 DEEPER DIVE: FAILURE ANALYSIS

In this subsection, we closely study two tasks requiring two distinct reasoning capabilities from LLM agents: (1) fine-grained data-dependent reasoning. (2) holistic understanding of a potentially domain-specific data lake.

**Challenge 1: Fine-grained data-dependent reasoning.**

Figure 5: Data snippets for study cases. Multiple water testing entries for each location may exist.

*environment-q17: What is the seasonal bacteria exceedance rate of Chatham's Bucks Creek Beach in the June, July, Aug of 2016? Impute missing values with median of the month in non-missing years.*

To solve this query, a correct pipeline must analyze the data present in both files in Figure 5. DS-Guru uses *OPS* sampling, which may not see or realize the "M" buried in the data and deduce that "M" stands for missing values. Although few-shot prompting enables the agent to see relevant errors, the lack of an explicit agentic control flow results in the LLM not connecting the execution errors to these fine-grained data observations. By contrast, on every agentic iteration, smolagents DR conjectures what the important data are to look at next to ensure the correctness of the pipeline it has drafted. This conjecture guides its tool call-enabled retrieval step. It subsequently analyzes the tool call and pipeline execution results before the next iteration. This explicit *retrieve-revise-repeat* pattern tightly couples error feedbacks with data retrieval, which helps address the fine-grained data-dependent reasoning challenge and leads to working end-to-end pipelines.

**Challenge 2: Holistic understanding of the input data and prior knowledge.**

*environment-q16: How many beaches remained safe to swimming from 2002 to 2023 inclusive?*

*environment-q16-3: How many beaches are there?*

*environment-q16-3* is an example sub-task for *environment-q16*, which also uses files in Figure 5. To solve the full task reliably, a system should be able to identify all beaches to start with. *environment-q16-3* prompts the system to carryout this identification and verifies the result.

The challenge with beach identification is that the "Beach Name" column encodes both the beach and sampling location. Cliff Pond (DCR) @ Main refers to the Main (street) sampling location of the Cliff

Pond beach (Figure 5). Facing many near-duplicate files in the data lake, systems do not have a clear global schema or geographical domain knowledge that they could use to understand this encoding scheme. As a result, both DS-Guru and smolagents DR failed on this sub-task, despite smolagents DR's agentic control flow. This case highlights the need to incorporate prior knowledge and discover clarifications about under-specified conventions *from the data* (Mao et al., 2019). Towards this end, we analyzed the traces of DS-Guru (few-shot, GPT-o3 & Claude 3.5) with the agentic system diagnosis framework proposed in Cemri et al. (2025). Respectively $24\%$ (GPT-o3) and $43\%$ (Claude 3.5) of all 104 tasks suffer from "failure to ask for clarification" and thus were not solved correctly. However, **KRAMABENCH** expects that a human expert could solve the tasks without additional clarifications by exploring and understanding the data. In this example, a human could decipher the beach name conventions with common U.S. geographical knowledge. Reasoning models currently lack similar capabilities to gain holistic understandings of the input data and fail to incorporate prior knowledge.

## 5 RELATED WORK

**LLM-Powered Agentic Systems.** There is a large and fast-growing literature on LLM-powered AI systems. These systems take on vastly different designs, such as vanilla LLM calls to frontier pre-trained reasoning models (OpenAI et al., 2024; DeepSeek-AI et al., 2025; Zhong et al., 2024), retrieval-augmented generation (Lewis et al., 2020), agentic workflow systems (Zhang et al., 2025b), chain-of-thought and iterative calls (Wei et al., 2022; Press et al., 2023), reflections (Ji et al., 2023) and task-time verifications (Tang et al., 2024a), structured knowledge representations (Jiang et al., 2024; Su et al., 2025; Wang et al., 2025), and data processing centric systems (Liu et al., 2024; Patel et al., 2025; Shankar et al., 2024). Recent work applies these techniques to data science tasks. For example, DocWrangler (Shankar et al., 2025) is an integrated development environment that helps the user optimize LLM prompts to construct data processing programs. DSAgent (Guo et al., 2024) is a framework that uses LLMs to understand user needs and build data science pipelines. Evaporate (Arora et al., 2023) helps users transform data into queryable tables. AutoPrep (Fan et al., 2025) constructs a data preparation program over a single table for a given question. Despite the progress, evaluating agent performance in real-world end-to-end setting remains a challenge.

**Evaluations of LLM-Powered Agentic Systems.** Benchmarks for question answering (QA) have shifted toward evaluating agentic solutions. These benchmarks require iterative retrieval, query parsing, planning, tool use, and temporal awareness. Recent works include FanOutQA (Zhu et al., 2024), MultiHop-RAG (Tang & Yang, 2024), CRAG (Yang et al., 2024), BrowseComp (Wei et al., 2025), which test end-to-end retrieval systems, MEQA (Li et al., 2024) for multi-hop reasoning with explanation chains, and MINTQA (He et al., 2024) for scaffolding long knowledge. These tasks differ from data science tasks, as they only require information retrieval and joins, but no data-intensive processing. Benchmarks such as DS-1000 (Lai et al., 2023), DA-Code (Huang et al., 2024), ARCADE (Yin et al., 2023), DataSciBench (Zhang et al., 2025a), DSEval (Zhang et al., 2024c) focus instead on implementing detailed instructions in general programming languages, specifically in data science tasks, differentiating themselves from other benchmarks like SWE-Bench (Jimenez et al., 2024), ML-Bench (Tang et al., 2024b), BigCodeBench (Zhuo et al., 2025). More recently, new benchmarks such as DSBench (Jing et al., 2025) and BLADE (Gu et al., 2024) have started to evaluate the ability to create an implementation plan. Benchmarks like ScienceAgentBench (Chen et al., 2025) and BixBench (Mitchener et al., 2025) evaluate using domain knowledge. Although such benchmarks assess specific capabilities, they fall short of capturing the full complexity of real-world data science pipelines.

## 6 CONCLUSION

**KRAMABENCH** evaluates the capabilities of systems to generate data science pipelines over a data lake consisting of heterogeneous, unclean input. Our comprehensive experiments using 8 LLMs across 4 different agentic systems with **KRAMABENCH** reveals although current systems are equipped with useful techniques such as agentic control flow and generic coding abilities, they are still far from solving real-world data science problems. Our analyses highlight several underexplored challenges such as effective retrieval, data-dependent reasoning, plan revision, and robust prior/domain knowledge integration as meaningful research directions towards practical automated data science systems.

## ETHICS STATEMENT

We acknowledge the limitations of **KRAMABENCH** regarding its scope, language, and cultural biases, and domain coverage, which stem from the human effort required for high-quality curation. All data included is publicly available, anonymized, or pseudonymized, with no personally identifiable information. The biomedical domain contains public data sourced from the cancer data commons (CDC) – this data is pseudonymized and does not require confidential access nor specific approvals, with the only sensitive attribute included as part of the workload being the pseudonymized age of patients. We emphasize privacy as paramount and warn users of the benchmark against potential identification risks, which we deem unlikely, associated with this data source. In future iterations of our benchmark we aim at broaden domain diversity, include multilingual data, and integrate community contributions to reduce existing biases Furthermore, we comply with licensing practices of data sources. For data sources that are publicly available but have redistribution constraints, we do not modify or separately host these datasets. Instead, we point users of our benchmark to the original data sources.

## REPRODUCIBILITY STATEMENT

We provide full artifacts—including code, data, workloads, and evaluation scripts—via our public repository at `https://github.com/mitdbg/Kramabench`. The main paper section 2 and Appendix D describe the process obtained to design and curate the task based on the datasets for each domain. Scripts to reproduce these steps can be found in the main repository. All datasets, benchmark frameworks, benchmark curation semi-automation scripts, reference pipelines and other accompanying annotations, and our reference system DS-Guru are available in our repository. The experimental analysis of different system under test in Section 4 can be reproduced using Python scripts also available in the public repository.

## ACKNOWLEDGMENTS

We are grateful for the support from the DARPA ASKEM Award HR00112220042, the ARPA-H Biomedical Data Fabric project, NSF DBI 2327954, a grant from Liberty Mutual, a Google Research Award, and the Amazon Research Award. Additionally, our work has been supported by contributions from Amazon, Google, and Intel as part of the MIT Data Systems and AI Lab (DSAIL) at MIT, along with NSF IIS 1900933. This research was sponsored by the United States Air Force Research Laboratory and the Department of the Air Force Artificial Intelligence Accelerator and was accomplished under Cooperative Agreement Number FA8750-19-2-1000. The views and conclusions contained in this document are those of the authors and should not be interpreted as representing the official policies, either expressed or implied, of the Department of the Air Force or the U.S. Government. The U.S. Government is authorized to reproduce and distribute reprints for Government purposes notwithstanding any copyright notation herein.

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

# A EXTENDED EXPERIMENT RESULTS

In this section, we supply the full evaluation results for which we presented a summary of in the main text due to space constraints. The DeepAnalyze system is from Zhang et al..

Table 9: Results by domain for **KRAMABENCH** on DS-Guru and smolagents DR with *Full* mode.

| System | Models | Archeology | Astronomy | Biomedical | Environment | Legal | Wildfire | Overall |
|---|---|---|---|---|---|---|---|---|
| DS-Guru no-context | GPT-o3 | 16.67 ± 00.00 | 0.00 ± 00.00 | 0.10 ± 0.14 | 5.00 ± 4.08 | 0.00 ± 00.00 | 14.18 ± 4.28 | 5.87 ± 0.71 |
| | GPT-4o | 0.00 ± 00.00 | 0.00 ± 00.00 | 0.00 ± 00.00 | 0.00 ± 00.00 | 0.00 ± 00.00 | 0.00 ± 00.00 | 0.00 ± 00.00 |
| | Llama-3.3-Instruct | 0.00 ± 00.00 | 0.00 ± 00.00 | 0.00 ± 00.00 | 0.00 ± 00.00 | 0.00 ± 00.00 | 1.59 ± 2.24 | 0.33 ± 0.46 |
| | Deepseek-R1 | 2.78 ± 3.93 | 0.00 ± 00.00 | 0.00 ± 00.00 | 0.00 ± 00.00 | 0.00 ± 00.00 | 0.00 ± 00.00 | 0.33 ± 0.46 |
| | Qwen2-5Coder* | 0.00 | 1.37 | 2.02 | 1.07 | 1.44 | 13.68 | 3.72 |
| | Claude-3.5* | 16.67 | 1.62 | 2.87 | 1.17 | 7.33 | 13.63 | 7.45 |
| | Claude-3.7 | 2.78 ± 3.93 | 3.33 ± 4.71 | 0.00 ± 00.00 | 0.00 ± 00.00 | 1.11 ± 1.57 | 4.38 ± 3.92 | 1.88 ± 0.81 |
| DS-Guru one-shot | GPT-o3 | 19.44 ± 3.93 | 0.00 ± 00.00 | 0.53 ± 0.75 | 16.11 ± 9.26 | 10.00 ± 00.00 | 40.01 ± 4.22 | 16.67 ± 2.93 |
| | GPT-4o | 8.33 ± 6.80 | 6.67 ± 4.71 | 3.70 ± 00.00 | 0.00 ± 00.00 | 4.41 ± 2.17 | 13.69 ± 3.06 | 6.08 ± 1.25 |
| | Llama-3.3-Instruct | 16.67 ± 00.00 | 0.00 ± 00.00 | 0.10 ± 0.14 | 0.00 ± 00.00 | 0.74 ± 0.52 | 5.97 ± 2.55 | 3.42 ± 0.37 |
| | Deepseek-R1 | 0.00 ± 00.00 | 0.00 ± 00.00 | 0.00 ± 00.00 | 0.00 ± 00.00 | 0.00 ± 00.00 | 0.00 ± 00.00 | 0.00 ± 00.00 |
| | Qwen2-5 Coder* | 0.00 | 1.36 | 2.22 | 12.59 | 1.15 | 16.48 | 6.43 |
| | Claude-3.5* | 0.00 | 4.15 | 2.15 | 6.21 | 6.68 | 34.99 | 10.85 |
| | Claude-3.7 | 2.78 ± 3.93 | 0.00 ± 00.00 | 0.00 ± 00.00 | 3.33 ± 2.36 | 0.00 ± 00.00 | 1.59 ± 2.24 | 1.31 ± 1.22 |
| DS-Guru few-shot | GPT-o3 | 13.89 ± 3.93 | 0.00 ± 00.00 | 0.10 ± 0.14 | 49.56 ± 0.87 | 9.26 ± 2.92 | 52.92 ± 0.72 | 24.98 ± 1.25 |
| | GPT-4o | 13.89 ± 3.93 | 3.33 ± 4.71 | 0.00 ± 00.00 | 0.00 ± 00.00 | 4.84 ± 3.16 | 25.69 ± 1.15 | 8.67 ± 1.48 |
| | Llama-3.3-Instruct | 22.22 ± 3.93 | 0.00 ± 00.00 | 0.20 ± 0.14 | 0.00 ± 00.00 | 3.33 ± 00.00 | 23.25 ± 3.06 | 8.40 ± 1.01 |
| | Deepseek-R1 | 0.00 ± 00.00 | 0.00 ± 00.00 | 0.00 ± 00.00 | 0.00 ± 00.00 | 0.00 ± 00.00 | 1.59 ± 2.24 | 0.33 ± 0.46 |
| | Qwen2-5Coder* | 8.33 | 2.40 | 4.35 | 12.64 | 9.06 | 16.48 | 9.98 |
| | Claude-3.5* | 16.67 | 1.52 | 1.96 | 11.21 | 7.01 | 39.16 | 14.35 |
| | Claude-3.7 | 5.56 ± 3.93 | 0.00 ± 00.00 | 0.00 ± 00.00 | 15.00 ± 4.08 | 8.89 ± 6.29 | 10.09 ± 9.67 | 8.29 ± 1.12 |
| smolagents DR | GPT-o3 | 8.33 ± 6.80 | 0.00 ± 00.00 | 0.00 ± 00.00 | 1.42 ± 1.09 | 4.51 ± 0.84 | 22.01 ± 1.39 | 7.57 ± 1.28 |
| | GPT-4o* | 33.33 | 0.00 | 11.11 | 35.00 | 40.00 | 38.10 | 30.77 |
| | Claude-3.5* | 33.33 | 0.00 | 22.22 | 60.00 | 46.67 | 52.38 | 41.35 |
| | Claude-3.7 | 44.44 ± 3.93 | 50.00 ± 8.16 | 38.89 ± 7.86 | 60.56 ± 7.74 | 61.23 ± 12.70 | 60.16 ± 7.43 | 55.83 ± 3.41 |
| smolagents Reflexion | GPT-o3 | 30.56 ± 10.39 | 46.67 ± 4.71 | 29.63 ± 5.24 | 23.89 ± 14.55 | 26.67 ± 14.40 | 35.04 ± 8.36 | 30.53 ± 10.79 |
| | Claude-3.7 | 41.67 ± 0.00 | 45.00 ± 5.00 | 50.00 ± 27.78 | 56.25 ± 8.75 | 58.33 ± 1.67 | 65.38 ± 0.86 | 55.37 ± 3.36 |
| smolagents PDT | GPT-o3 | 8.33 ± 6.80 | 0.00 ± 00.00 | 0.00 ± 00.00 | 1.42 ± 1.09 | 4.51 ± 0.84 | 22.01 ± 1.39 | 7.57 ± 1.28 |
| | Claude-3.7 | 19.44 ± 7.86 | 6.67 ± 4.71 | 5.56 ± 5.56 | 9.84 ± 2.86 | 6.25 ± 4.95 | 22.58 ± 4.05 | 12.01 ± 1.03 |
| DeepAnalyze* | DeepAnalyze-8B | 33.33 | 16.67 | 26.67 | 24.67 | 18.45 | 19.61 | 14.98 |

Table 10: Results by domain for **KRAMABENCH** on DS-Guru and smolagents DR with *Oracle* mode.

| System | Models | Archeology | Astronomy | Biomedical | Environment | Legal | Wildfire | Total |
|---|---|---|---|---|---|---|---|---|
| DS-Guru no-context | GPT-o3 | 11.11 ± 3.93 | 10.00 ± 8.16 | 11.11 ± 0.00 | 0.00 ± 0.00 | 0.00 ± 0.00 | 10.17 ± 0.92 | 5.36 ± 1.20 |
| | GPT-4o | 0.00 ± 0.00 | 0.00 ± 0.00 | 0.00 ± 0.00 | 0.00 ± 0.00 | 0.00 ± 0.00 | 0.00 ± 0.00 | 0.00 ± 0.00 |
| | Llama-3.3-Instruct | 5.56 ± 3.93 | 0.00 ± 0.00 | 11.11 ± 0.00 | 0.00 ± 0.00 | 0.00 ± 0.00 | 1.59 ± 2.24 | 1.96 ± 0.80 |
| | Deepseek-R1 | 0.00 ± 0.00 | 0.00 ± 0.00 | 0.00 ± 0.00 | 0.00 ± 0.00 | 0.00 ± 0.00 | 0.00 ± 0.00 | 0.00 ± 0.00 |
| | Qwen2-5Coder* | 10.24 | 6.74 | 7.71 | 7.14 | 1.52 | 4.53 | 6.62 |
| | Claude-3.5* | 16.52 | 10.63 | 9.87 | 12.51 | 9.80 | 0.00 | 11.63 |
| | Claude-3.7 | 4.17 ± 4.17 | 10.00 ± 0.00 | 0.00 ± 0.00 | 1.58 ± 1.58 | 1.67 ± 1.67 | 8.96 ± 0.57 | 4.11 ± 0.79 |
| DS-Guru one-shot | GPT-o3 | 8.33 ± 6.80 | 16.67 ± 4.71 | 7.51 ± 5.31 | 23.89 ± 6.71 | 15.56 ± 1.57 | 42.80 ± 3.48 | 21.35 ± 0.84 |
| | GPT-4o | 11.11 ± 7.86 | 13.33 ± 4.71 | 5.19 ± 7.33 | 12.33 ± 3.30 | 6.67 ± 0.00 | 19.85 ± 3.61 | 11.54 ± 1.47 |
| | Llama-3.3-Instruct | 0.00 ± 0.00 | 6.67 ± 4.71 | 7.41 ± 5.24 | 0.00 ± 0.00 | 0.00 ± 0.00 | 26.37 ± 5.02 | 6.74 ± 1.49 |
| | Deepseek-R1 | 2.78 ± 3.93 | 6.67 ± 4.71 | 0.00 ± 0.00 | 0.00 ± 0.00 | 0.00 ± 0.00 | 0.00 ± 0.00 | 0.98 ± 0.80 |
| | Qwen2-5Coder* | 9.72 | 11.57 | 5.37 | 15.13 | 8.96 | 13.22 | 11.26 |
| | Claude-3.5* | 17.07 | 10.24 | 9.44 | 22.27 | 11.47 | 17.93 | 15.48 |
| | Claude-3.7 | 0.00 ± 0.00 | 0.00 ± 0.00 | 0.00 ± 0.00 | 6.67 ± 1.67 | 3.33 ± 3.33 | 4.76 ± 0.00 | 3.27 ± 1.31 |
| DS-Guru few-shot | GPT-o3 | 16.67 ± 0.00 | 26.67 ± 4.71 | 7.41 ± 5.24 | 50.11 ± 4.09 | 61.03 ± 5.78 | 52.02 ± 3.79 | 43.71 ± 1.94 |
| | GPT-4o | 13.89 ± 3.93 | 20.00 ± 0.00 | 3.70 ± 5.24 | 18.44 ± 6.00 | 37.78 ± 4.16 | 36.67 ± 3.53 | 26.20 ± 2.34 |
| | Llama-3.3-Instruct | 13.89 ± 3.93 | 10.00 ± 0.00 | 11.11 ± 0.00 | 0.00 ± 0.00 | 6.67 ± 0.00 | 29.72 ± 1.27 | 11.67 ± 0.65 |
| | Deepseek-R1 | 5.56 ± 3.93 | 6.67 ± 4.71 | 0.00 ± 0.00 | 0.00 ± 0.00 | 0.00 ± 0.00 | 0.00 ± 0.00 | 1.31 ± 0.46 |
| | Qwen2-5Coder* | 11.83 | 14.91 | 7.51 | 18.39 | 13.70 | 18.51 | 15.15 |
| | Claude-3.5* | 16.24 | 14.02 | 14.80 | 33.83 | 26.36 | 25.02 | 24.22 |
| | Claude-3.7 | 20.83 ± 4.17 | 25.00 ± 15.00 | 0.00 ± 0.00 | 9.17 ± 9.17 | 31.67 ± 1.67 | 18.17 ± 5.20 | 19.75 ± 3.36 |
| smolagents DR | GPT-o3 | 27.78 ± 7.86 | 26.67 ± 4.71 | 37.04 ± 5.24 | 24.00 ± 0.00 | 17.78 ± 5.67 | 32.89 ± 10.31 | 25.86 ± 4.57 |
| | GPT-4o* | 25.00 | 25.00 | 22.22 | 20.00 | 56.67 | 38.10 | 39.00 |
| | Claude-3.5* | 16.67 | 25.00 | 33.33 | 25.00 | 66.66 | 66.66 | 47.00 |
| | Claude-3.7 | 45.83 ± 12.50 | 65.00 ± 5.00 | 58.33 ± 2.78 | 46.83 ± 1.50 | 65.00 ± 1.67 | 75.08 ± 3.78 | 60.67 ± 1.23 |
| smolagents Reflexion | GPT-o3 | 30.56 ± 3.93 | 40.00 ± 0.00 | 44.44 ± 0.00 | 31.44 ± 2.20 | 22.22 ± 8.31 | 39.78 ± 6.14 | 32.33 ± 4.44 |
| | Claude-3.7 | 37.50 ± 4.17 | 50.00 ± 0.00 | 83.33 ± 5.56 | 62.67 ± 4.33 | 70.00 ± 10.00 | 64.45 ± 1.39 | 62.81 ± 3.10 |
| smolagents PDT | GPT-o3 | 8.33 ± 6.80 | 0.00 ± 0.00 | 0.00 ± 0.00 | 3.26 ± 1.30 | 8.62 ± 3.88 | 15.86 ± 5.53 | 7.69 ± 2.69 |
| | Claude-3.7 | 16.67 ± 0.00 | 10.00 ± 0.00 | 11.11 ± 0.00 | 11.50 ± 3.50 | 9.24 ± 0.35 | 25.50 ± 5.46 | 14.38 ± 2.15 |

# B DS-GURU DETAILS

The baseline system we provide, DS-Guru, follows a simple design. For each task, the system provides the backend LLM with an informative sample of data from each file in the data lake first as well as the task prompt. DS-Guru leverages instruction tuning to guide the LLM backend to provide

Table 11: Results by domain for **KRAMABENCH** (*Trimmed* input lake). ⋆ marks web-browser on.

| System | Metric | Domains | | | | | | |
| --- | --- | --- | --- | --- | --- | --- | --- | --- |
| | | Archeology | Astronomy | Biomedical | Environment | Legal | Wildfire | **Total** |
| DS-Guru | Score | 25.00 ± 6.80 | 0.00 ± 0.00 | 3.70 ± 5.24 | 48.33 ± 8.50 | 51.11 ± 3.14 | 48.87 ± 7.90 | 37.84 ± 3.90 |
| few-shot (GPTo3) | Avg. runtime/task (min) | 2.8502 | **1.6584** | 14.4576 | **2.3543** | 3.8403 | 2.3559 | 3.8076 |
| smolagents DR | Score | **50.00 ± 0.00** | **80.00 ± 0.00** | 55.56 ± 0.00 | 40.83 ± 0.00 | **60.00 ± 0.00** | 67.24 ± 0.00 | **58.12 ± 0.00** |
| | Avg. runtime/task (min) | 2.3538 | 5.4944 | 17.9464 | 3.6673 | 3.4209 | 3.2526 | 4.8074 |
| smolagents Reflexion | Score | 33.33 ± 0.00 | 50.00 ± 0.00 | **66.67 ± 0.00** | 43.33 ± 0.00 | 55.54 ± 0.00 | 64.44 ± 0.00 | 52.81 ± 0.00 |
| | Avg. runtime/task (min) | 6.451 | 7.2277 | 43.216 | 7.3128 | 6.5276 | 5.3909 | 9.696 |
| smolagents PDT | Score | 16.67 ± 0.00 | 0.00 ± 0.00 | 0.00 ± 0.00 | 14.37 ± 0.00 | 13.33 ± 0.00 | 0.00 ± 0.00 | 8.70 ± 0.00 |
| | Avg. runtime/task (min) | 7.2708 | 6.3041 | 15.5661 | 6.6759 | 8.031 | 8.5672 | 8.2438 |
| OpenAI DR⋆ | Score | 40.00 | 33.33 | 44.45 | **61.67** | 50.00 | **67.28** | 52.18 |
| | Avg. runtime/task (min) | 8.105 | 20.16 | 10.67 | 5.3 | 8.68 | 12.62 | 10.35 |
| Gemini 2.5 Pro⋆ | Score | 25.00 | 16.67 | 33.33 | 25.00 | 13.33 | 24.87 | 18.48 |
| | Avg. runtime/task (min) | **0.64** | 2.44 | **3.49** | 2.3975 | **3.105** | **2.314** | **2.4835** |

Table 12: Cost-accuracy Tradeoff between different SUTs under *Full* input mode.

| SUT | Overall Accuracy | Overall Runtime | Accuracy/ Runtime | Accuracy / 1k In Tokens | Accuracy / 1k Out Tokens |
| --- | --- | --- | --- | --- | --- |
| GPTo3 - Naive | 05.87 ± 00.71 | 01h 01m 41s ± 06m 26s | 00.17 ± 00.02 | 04.73 ± 00.51 | 02.69 ± 00.31 |
| GPTo3 - One Shot | 16.67 ± 02.93 | 01h 11m 26s ± 08m 13s | 00.42 ± 00.12 | 00.43 ± 00.07 | 09.16 ± 01.63 |
| GPTo3 - Few Shot | 24.98 ± 01.25 | 02h 30m 42s ± 42m 34s | 00.31 ± 00.07 | 00.29 ± 00.05 | 08.58 ± 00.61 |
| GPT4o - Naive | 00.00 ± 00.00 | 38m 42s ± 02m 23s | 00.00 ± 00.00 | 00.00 ± 00.00 | 00.00 ± 00.00 |
| GPT4o - One Shot | 06.08 ± 01.25 | 39m 15s ± 02m 11s | 00.27 ± 00.06 | 00.45 ± 00.09 | 09.81 ± 02.06 |
| GPT4o - Few Shot | 08.67 ± 01.48 | 01h 13m 13s ± 18m 12s | 00.22 ± 00.05 | 00.26 ± 00.03 | 07.68 ± 01.33 |
| Llama3 Instruct - Naive | 00.33 ± 00.46 | 35m 09s ± 03m 53s | 00.01 ± 00.02 | 00.26 ± 00.37 | 00.37 ± 00.52 |
| Llama3 Instruct - One Shot | 03.42 ± 00.37 | 01h 38m 51s ± 11m 06s | 00.06 ± 00.01 | 00.25 ± 00.03 | 05.63 ± 00.61 |
| Llama3 Instruct - Few Shot | 08.40 ± 01.01 | 02h 10m 11s ± 21m 06s | 00.12 ± 00.03 | 00.27 ± 00.04 | 08.98 ± 01.20 |
| DeepseekR1 - Naive | 00.33 ± 00.46 | 01h 36m 29s ± 07m 12s | 00.01 ± 00.01 | 00.26 ± 00.37 | 00.16 ± 00.23 |
| DeepseekR1 - One Shot | 00.00 ± 00.00 | 01h 46m 09s ± 04m 42s | 00.00 ± 00.00 | 00.00 ± 00.00 | 00.00 ± 00.00 |
| DeepseekR1 - Few Shot | 00.33 ± 00.46 | 01h 56m 18s ± 12m 06s | 00.01 ± 00.01 | 00.01 ± 00.01 | 00.15 ± 00.22 |

a Python implementation of the task pipeline as well as a structured explanation of the steps to be taken. DS-Guru then executes the implementation and iterate with the LLM pipeline to debug and improve the pipeline by supplying outputs and error messages.

The prompt used to instruct the LLM backend to provide a pipeline for the end-to-end task is presented below:

## B.1 SYSTEM PROMPT

```
You are a helpful assistant that generates a plan to solve
the given request, and you'll be given:Your task is to answer
the following question based on the provided data sources.
Question: {query}
Data file names: {file_names}
The following is a snippet of the data files: {data}
Now think step-by-step carefully.
First, provide a step-by-step reasoning of how you would arrive
at the correct answer.
Do not assume the data files are clean or well-structured
(e.g., missing values, inconsistent data type in a column).
Do not assume the data type of the columns is what you see in
the data snippet (e.g., 2012 in Year could be a string, instead
of an int). So you need to convert it to the correct type if
your subsequent code relies on the correct data type (e.g.,
cast two columns to the same type before joining the two
tables).
You have to consider the possible data issues observed in the
data snippet and how to handle them.
```

```
Output the steps in a JSON format with the following keys:
- id: always "main-task" for the main task. For each subtask,
use "subtask-1", "subtask-2", etc.
- query: the question the step is trying to answer. Copy down
the question from above for the main task.
- data_sources: the data sources you need to check to answer
the question. Include all the file names you need for the main
task.
- subtasks: a list of subtasks. Each subtask should have the
same structure as the main task.
For example, a JSON object for the task might look like this:
{example_json}
You can have multiple steps, and each step should be a JSON
object. Your output for this task should be a JSON array of
JSON objects.
Mark the JSON array with {json_notation} to indicate the start
and end of the code block.
Then, provide the corresponding Python code to extract the
answer from the data sources.
The data sources you may need to answer the question are:
{file_paths}.
If possible, print the answer (in a JSON format) to each step
you provided in the JSON array using the print() function.
Use "id" as the key to print the answer.
For example, if you have an answer to subtask-1, subtask-2, and
main-task (i.e., the final answer), you should print it like
this:
print(json.dumps(
{{"subtask-1": answer1,
"subtask-2": answer2,
"main-task": answer
}}, indent=4))
You can find a suitable indentation for the print statement.
Always import json at the beginning of your code.
Mark the code with {notations} to indicate the start and end of
the code block.
```

## B.2 ABLATION STUDIES

We have started conducted ablation studies on key hyper-parameters, using the best-performing configuration of DS-Guru (i.e., self-correcting with GPT-o3). Here are our preliminary findings: The quality performance is positively correlated to token usage [1]. When varying the number of rows sampled per table, our result is consistent — success goes up as we sample more rows. We then observed a decrease at n=100, which is caused by the limited context window and our naive sampling algorithm. DS-Guru falls back to no data snippet when the prompt exceeds the context limit. DS-Guru showed consistent success across different numbers of maximum tries, with an initial slight increase. This potentially has two implications: (i) compile/runtime errors are not the major cause of failures; (ii) in a single-agent system, it may be difficult for the agent to get unstuck from a loop when fixing the error. We will discuss this in depth in failure analysis. We will update the paper to present these results and discuss them analytically under our 3-level evaluation framework. For reference, the full table of results is as follows: Varying the number of rows sampled in the input data snippet.

## C SMOLAGENTS AGENTIC BASELINE DETAILS

In this section, we describe the single-agent and multi-agent baselines systems we evaluated on **KRAMABENCH** more.

Table 13: DS-Guru with GPT-o3: performance and cost across different numbers of iterations.

| Number of Iterations | 5 | 10 | 15 | 20 |
|---|---|---|---|---|
| Overall Performance (%) | 23.36 | 22.83 | 20.73 | 21.33 |
| Tokens/Iteration (Mean) | 64,548.9 | 72,926.3 | 70,845.1 | 72,301.7 |

Table 14: Runtime performance by number of sampled rows per file. Runtime is in seconds.

| SUT | Archeology | Astronomy | Biomedical | Environment | Legal | Wildfire | Overall | Runtime |
|---|---|---|---|---|---|---|---|---|
| 10 Rows | 18.75 | 12.80 | 8.63 | 34.52 | 13.32 | 37.42 | 22.89 | 732.45 |
| 50 Rows | 23.48 | 10.55 | 7.87 | 37.60 | 14.08 | 40.63 | 24.68 | 655.61 |
| 100 Rows | 20.61 | 11.95 | 8.53 | 34.84 | 12.20 | 40.60 | 23.36 | 1374.82 |
| 150 Rows | 21.08 | 10.58 | 8.64 | 31.68 | 13.09 | 39.22 | 22.58 | 802.90 |

## C.1 SMOLAGENTS DR

For the single-agent baseline, we use the open source deep research implementation by `smolagents` (git). This agentic framework follows a canonical think → action → response loop with agentic actions expressed in code. In addition to code, the system is equipped with a text inspector capable of processing different common formats originally released with Microsoft Magentic One (mic). While the official implementation also equips the system with a web browser by default, we disabled the internet access to allow for direct comparison with DS-Guru.

## C.2 SMOLAGENTS REFLEXION

Our first multi-agent baseline is Reflexion (Shinn et al., 2023). In addition to an *Actor* agent, Reflexion (Figure 6) introduces (1) An *Evaluator* agent which provides internal feedback by evaluating the outcome of each action. (2)A *Self-reflection* agent which provides external feedback with the outcome and the evaluation of the action. Compared to traditional reinforcement learning techniques, feedback in Reflexion are expressed with natural language and stored in agent memory to guide future actions.

## C.3 SMOLAGENTS PDT

Our second multi-agent baseline is based on AutoPrep (Fan et al., 2025), a framework for natural language question answering over tabular data. We augmented AutoPrep with tools for parsing non-tabular data and the hierarchical task decomposition technique similar with DS-Guru to address the complexity of **KRAMABENCH** tasks. The original AutoPrep pipeline involves three agents: (1) Planner (2) Programmer (3) Executor. With the augmentations we implemented, the system employs two agents respectively playing the roles of (1) *Planner* and *(task) Decomposer* (3) *Tool Executor*. We implemented this approach also using `smolagents` and illustrate the architecture in Figure 7.

# D DATASET DETAILS

The six input domains with the associated studies that we used to design our benchmark tasks are:

- **Archeology**: the data files consists of chronological, archaeological, faunal, and botanical data supporting the presence of Holocene hunter-gatherers on the Maltese Islands in the Mediterranean from roughly 8000 years ago to 7500 years ago. The files were collected from the publicly available data associated with the papers Groucutt et al. (2021); Scerri et al. (2025).
- **Astronomy**: the data files consist of the OMNI dataset Papitashvili & King (2020a;b) that contains near-Earth solar wind, plasma, and magnetic field data, the Swarm dataset Siemes et al. (2016); European Space Agency (2013) that contains the magnetic field and geomagnetic field data, the SILSO Sunspot Number data Clette & Lefèvre (2015), Space-Track.org Two-Line Element Sets (TLEs) U.S. Space Command (2025), the National Oceanic and Atmospheric Administration (NOAA) Flux Forecast dataset U.S. Air Force & NOAA Space Weather Prediction Center (2025), and NOAA GOES Satellite dataset NOAA Office of Satellite and Product Operations (1994). The

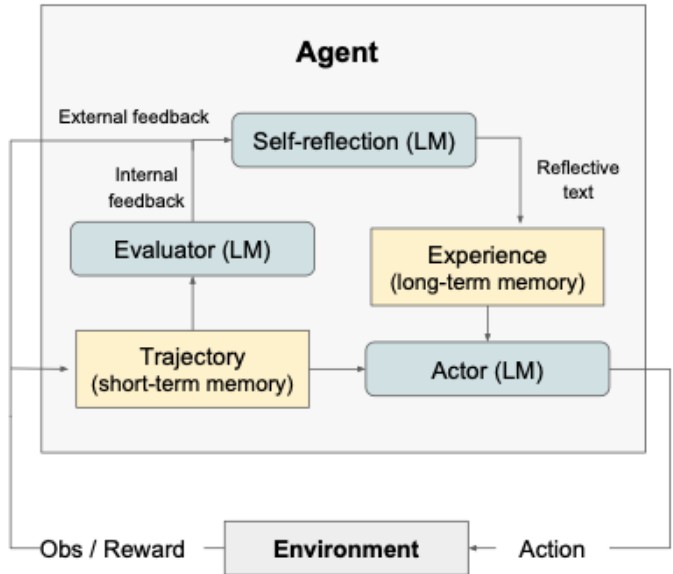

Figure 6: Architecture diagram of Reflexion. Reproduced from Figure 2(a) in Shinn et al. (2023).

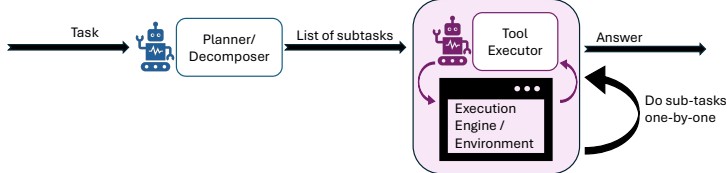

Figure 7: Architecture diagram of `smolagents-pdt`

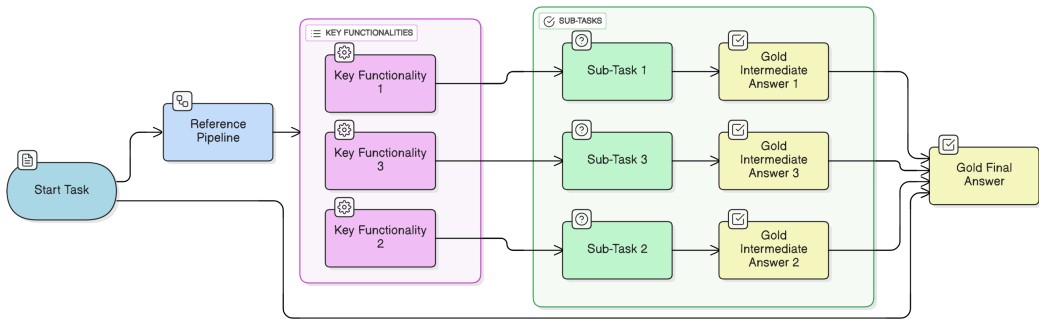

Figure 8: KRAMABENCH maps system evaluation to both pipeline-design and sub-task-level evaluations, enabling analysis of why models succeed or fail beyond end-to-end performance.

Table 15: Performance by number of tries. Runtime is in seconds.

| SUT | Archeology | Astronomy | Biomedical | Environment | Legal | Wildfire | Overall | Runtime |
|-----|-----------|-----------|-----------|-------------|-------|----------|---------|---------|
| 5 Tries | 20.61 | 11.95 | 8.53 | 34.84 | 12.20 | 40.60 | 23.36 | 1374.82 |
| 10 Tries | 19.86 | 11.60 | 8.71 | 36.66 | 10.79 | 37.86 | 22.83 | 575.88 |
| 15 Tries | 20.47 | 7.00 | 8.72 | 36.84 | 9.51 | 31.47 | 20.73 | 721.95 |

combination of these datasets has been used to analyze how activity from the Sun affects Earth's atmosphere, ocean currents, and weather by the authors of Briden et al. (2023); Parker & Linares (2024).

- **Biomedical**: the data files consist of the prote-ogenomic characterization of 95 prospectively collected endometrial carcinomas, respectively for 83 endometrioid and 12 serous tumors. Extensive analysis are done on these datasets to understand proteomic markers of tumor subgroups and regulatory mechanisms in the papers Dou et al. (2020); Gillette et al. (2020).
- **Environment**: the data files consist of beach water quality dataset from Massachusetts Environment Public Health Tracking (EPHT) Massachusetts Department of Public Health (2025b), the Massachusetts Bay beach dataset from Massachusetts Water Resources Authority (MWRA) Massachusetts Water Resources Authority (2025b), and the rainfall dataset from NOAA National Weather Service National Weather Service (2025), from 2002 to 2025. The data has been used in yearly reports Massachusetts Department of Public Health (2025a); Massachusetts Water Resources Authority (2025a) to uncover trends in beach water pollution and the correlation between rainfall and water quality.
- **Legal**: the datasets consists of 136 data files, accessible through the Federal Trade Commission (FTC) portal Federal Trade Commission (2025b) and Wikipedia Wikipedia contributors (2025), including information on merger filings, civil penalty actions, etc. The data is used in visualizations and dashboards that analyze nation-level debt collection and fraud detection, available at Federal Trade Commission (2025c;a).
- **Wildfire**: the datasets consists of NOAA wildfire dataset National Centers for Environmental Information (NCEI) (2025), National Interagency Fire Center (NIFC) Fire Information National Interagency Fire Center (2025), US Environmental Protection Agency (EPA) Air Quality Annual Data U.S. Environmental Protection Agency (2025), US Election 2020 Dataset Fontes (2020), Zillow Home Value Index Dataset Robikscube (2021), US Census 2020 U.S. Census Bureau (2025), and the Large wildfire Incident Status Summary Young et al. (2021) to understand wildfire incident location, cause, and consequences in the US from 2002 to 2016. This data has been used for analysis in the reports published by the NOAA and NIFC NCEI.Monitoring.Info@noaa.gov (2025); Center .

## D.1 Cross-domain Accuracy Difference Analysis

In this subsection, we discuss the findings on likely causes of the differences in accuracy between different domains in **KramaBench**. We obtained these findings by manually analyzing the traces of smolagents-reflexion DR.

1. Archeology (33.33%) : In this domain, the system correctly solves questions answerable from a single table. However, errors occur for tasks requiring joining tables found in different files, because it treats multiple files as raw text instead of loading them as tables.

2. Astronomy (16.67%) : Astronomy tasks have the lowest average performance. In this domain, a large portion of the required input data is found in proprietary scientific formats (e.g., FORTRAN-style dat files). We observed that the agent struggles whenever it needed to load data from these files, e.g., SP3 orbit files or satellite products.

3. Biomedical (44.44%) : When working with biomedical data, the agent is reliable for shallow operations on a single sheet but fails to navigate large, multi-sheet workbooks and join data across sheets. Cross-sheet joins, especially between clinical and phosphoproteomics data, are problematic, and errors arise in correlation statistics due to sign miscalculations.

4. Environment (60.00%) : In the environmental domain, the system performs well on the relatively tasks involving clean CSV data, such as filtering, counting, and averaging. Unlike other tasks which struggle from data retrieval or understanding issues, the main issues arise

Table 16: Detailed breakdown of per-domain tasks in KRAMABENCH. Reproduced from Table 3

| Domain | # tasks | # subtasks | % Hard Tasks | # datasets | # sources | File size |
|---|---|---|---|---|---|---|
| Archeology | 12 | 71 | 50.00% | 5 | 2 | 7.5MB |
| Astronomy | 12 | 68 | 50.00% | 1556 | 8 | 486MB |
| Biomedical | 9 | 38 | 66.66% | 7 | 2 | 175MB |
| Environment | 20 | 148 | 70.00% | 37 | 3 | 31MB |
| Legal | 30 | 188 | 53.33% | 136 | 2 | 1.3MB |
| Wildfire | 21 | 120 | 71.42% | 23 | 7 | 1GB |
| **Total** | 104 | 633 | 60.58% | 1764 | 24 | 1.7GB |

from implementation/arithmetic mistakes, such as incorrect aggregation scopes or rounding errors, which explains the higher score.

5. Legal (63.33%): In these tasks, the agent handles the straightforward pipelines well but struggles with loading data from messy files, i.e., that contain multi-row headers, partial subtotals, and metadata rows. Amongst the common errors that stem from these shortcomings, one example is that sum, means, and aggregations are only partial due to incorrect loading.

6. Wildfire (52.38%) : Within wildfire-related tasks, the system faces challenges with geospatial data and temporal/statistical reasoning. It struggles with GeoPackage layers and spatial joins, as well as with rolling-window aggregations for weather. However, text lookups and simple value comparisons work relatively well.

# E  TASK DETAILS

Across the 6 workloads, we supply 104 end-to-end data science pipelines. The table for the overall breakdown of the tasks over the workloads is reproduced at Table 16 for convenience. In this section, we use an example from the **archeology** workload to explain the organization of tasks.

Each workload is associated with a data lake consisting of tabular data and unstructured textual data.

```
archeology/input/:
    climateMeasurements.xlsx
    conflict_brecke.csv
    radiocarbon_database_regional.xlsx
    roman_cities.csv
    worldcities.csv
```

Before tasks in a workload are sent to the system under test, the system receives the directory where the data lake resides and may index it offline. When tasks are prompted, the system should not receive information on which files in the data lake the task pertains to. Each end-to-end task is specified with a high-level natural language prompt. Consider the following example of end-to-end task from the **archeology** domain:

```
 What is the average Potassium in ppm from the first and last
time the study recorded people in the Maltese area?  Assume
that Potassium is linearly interpolated between samples.
Round your answer to 4 decimal places.
```

For evaluating the performance of our systems, we use three artifacts:

1. The end-to-end ground truth answer used to calculate the overall end-to-end score.
2. A sequence of key functionalities, extracted from a manually verified reference implementation for the solution in Python.
3. A sequence of subtasks, natural language questions whose correct answer depends on correct code implementation of a key functionality.

The key functionalities are manually refined to correspond to the functionalities that should exist in any pipeline that produces the correct output. The sequence of key functionalities for the example end-to-end task above is the following:

```
1. Load the radiocarbon_database_regional.xlsx and
   climateMeasurements.xlsx and read the first worksheet
   of each.

2. Remove rows or columns that are entirely NaN or do not
   contain relevant information from both dataframes to
   ensure clean numeric processing.

3. Convert both chronologies to calendar years:  for the
   radio-carbon table get the year as 1950 minus the
   'date'

4. Convert both chronologies to calendar years:  for the
   climate table get the year as 1950 minus the rounded
   'Age_ky.1' (in thousands of years) multiplied by 1000.

5. Determine the span of human presence in the Maltese
   area by taking the minimum and maximum 'year' in the
   radio-carbon dataframe.

6. For every integer year within the human presence
   span, locate the closest earlier and later rows in the
   climate dataframe and linearly interpolate (or directly
   return) the Potassium value 'K' and collect all these
   values.

7. Compute the mean of the collected Potassium values.
```

For each key functionality, we supply a **subtask** associated with the key functionality. Each subtask is annotated with the ground truth subtask answer. These subtasks are used to verify the code implementation capabilities of systems under test. Note that among correct pipeline implementations for the end-to-end task, key functionalities may be ordered or composed differently. The subtasks associated to the end-to-end example task are:

```
1. Which files contain information about Potassium in ppm
   and the maltese people?

2. What are the indices (0-indexed) in rows in the climate
   measurement dataframe that must be cleaned?

3. What are the calendar years in the radiocarbon table?

4. What are the calendar years in the climate table?

5. What are the minimum and maximum years of radiocarbon
   dating for the Malta region?

6. What are the Potassium values for each integer year
   between -7580 and -4050 (included)?  If the value is
   not available, use interpolatation between the closest
   earlier and later values.

7. What is the mean potassium value for the years between
   -4462 and -4055?  Use 4 decimal places.
```

## F  HUMAN BASELINE DETAILS

In this section, we summarize how we conducted the human baseline and discuss the results and implications.

To contextualize LLM performance on KRAMABENCH, we conducted a human data science study involving nine participants. Each participant was assigned a subset of benchmark tasks and asked to solve them under the same data directory structure, resource constraints, and assumptions provided to our LLM agents. For every assigned task, participants produced:

- a complete, reproducible end-to-end solution in a Jupyter notebook,

- a detailed log of their active time, broken down into data exploration, pipeline design, coding, and debugging, and

- both draft-stage notes and a final clean solution, enabling direct comparison to LLM work-flows and error modes.

Participants followed standardized instructions covering repository setup, task management, time tracking, and reproducibility requirements. Across all human-authored solutions, we performed a detailed analysis of the underlying failure causes. The distribution of error types is:

- **Incorrect pipeline design** (46%): The largest category. These errors occur when, for example, experts mis-specified a join, aggregation rule, grouping key, or filtering logic. This suggests that the most cognitively demanding part of real-world data science is pipeline design, rather than implementation.

- **Lack of domain knowledge** (24%): Many tasks contain implicit domain assumptions (e.g., definitions of "violation," mapping categorical labels). Experts often produced internally consistent but mismatched interpretations. This shows that even humans struggle with domain-specific task semantics.

- **Incorrect inputs** (12%): Tasks often require gathering information across multiple similarly named or structurally similar files, and even humans sometimes use wrong inputs. These errors reflect the challenge of navigating multi-file datasets.

- **Incorrect answer format** (9%): Some errors are due to having the final outputs in the wrong representation (e.g., units, rounding, formatting), which did not match the one requested by the task.

- **Library/version issues** (9%): Minor inconsistencies (e.g., pandas handling) that changed intermediate results enough to fail strict correctness checking.

**Interpretation and implications.** Multi-file, multi-step pipelines are inherently error-prone—even for trained experts. Humans have difficulty navigating a vast data lake, which we see as an opportunity for LLM-powered systems to quickly search through the lake and identify the target files. Having a reliable retriever could greatly improve accuracy. Ambiguity and assumed domain knowledge are a real factor in real-world data tasks. One possible way for future agentic data-science systems to combat this issue is to ask clarification questions and invite user input. Another approach is to branch out on possible solutions by clearly stating the assumptions. Pipeline design is the bottleneck. Nearly half of all errors (45.45%) are due to incorrect pipeline logic, highlighting that the core challenge is understanding what transformations to perform, not coding them. Overall, most human errors stemmed from misinterpreting ambiguous tasks, selecting the wrong files, or designing incorrect pipelines—challenges that mirror the dominant failure modes of LLM agents. This confirms that KRAMABENCH captures genuinely difficult, real-world data-to-insight tasks where even trained data scientists struggle with pipeline reasoning, multi-file navigation, and implicit domain assumptions.

## G    EVALUATION DETAILS

Considering the broad nature of data science tasks, and the challenges in correctly evaluating their design and implementation, **KRAMABENCH** evaluates systems on three capabilities. From the most to the least automated: (1) End-to-end automation (2) Pipeline design (3) Sub-task implementation.

We are primarily interested in systems that can solve end-to-end data science tasks fully correctly, which drives our main evaluation metric to be the result from the end-to-end automation setting.

Table 17: Answer type and example questions

| Type | Example | Metric | Scoring |
|---|---|---|---|
| String (exact) | The name of a file to load. | Accuracy | 0/1 |
| String (approximate) | The month when an experiment started. | ParaPluie paraphrase detection (Lemesle et al., 2025) | 0/1 |
| Numeric (exact) | Counting the number of entries satisfying a predicate. | Accuracy | 0/1 |
| Numeric (approximate) | Prediction of a future experiment observation. | Relative Absolute Error (RAE) $|\hat{y} - y|/|y|$ | $1/(1 + \mathrm{RAE})$ |
| List (exact) | Names of columns to filter data. | F1 (exact match) | F1 score |
| List (approximate) | Regression coefficients for different variables. | F1 score (approximate match > 0.9) | F1 score |

## G.1 MAIN METRIC: END-TO-END AUTOMATION SETTING

Each task in **KRAMABENCH** has a manually validated target output and is scored from [0,1]. Since pipelines might be composed of steps with varying nature, we identify six possible answer types for the target output. summarized and discussed in Table 3. For each answer type, we choose a scoring scheme normalized to the range $[0, 1]$, also shown in Table 3. When tested, the **total** score of system $F$ for a workload $W$ is defined solely based on the end-to-end correctness as

$$\frac{\sum_{T \in W} \mathrm{score}(F(T))}{|W|}$$

Each $T$ is a task belonging to workload $W$, and $|W|$ is the number of tasks in workload $W$. The overall score for the entire benchmark suite is defined analogously.

## G.2 LLM-AS-A-JUDGE VALIDATION

To assess the validity of the evaluation for `String (approximate)` and `List (approximate)` with `String (approximate)` list members conducted via instruction tuning an LLM, we performed a small scale human-LLM evaluator agreement study. We asked three human reviewers to manually evaluate the equivalence between the reference solutions and the answers generated by 12 different SUTs (the three variants of DS-Guru across four different LLM backends). We run the LLM-as-a-judge evaluation pipeline three times. We report the Cohen's Kappa values for inter-human agreement, human-LLM agreement and inter-LLM calibration (Table 18). The possible values range from -1 (complete misalignment) to 1 (complete alignment). The results show very high inter-human agreement ($\sim 95\%$ on average) and moderately high human-LLM agreement ($\sim 84\%$ on average), indicating that our usage of LLM-as-a-judge provides meaningful evaluation results.

Table 18: Inter-Human, inter-LLM, and human-LLM agreement on approximate answer evaluation.

| Inter-Human Agreement | | | Inter-LLM Agreement | | | Human–LLM Agreement | | |
|---|---|---|---|---|---|---|---|---|
| **Rater 1** | **Rater 2** | $\kappa$ | **Rater 1** | **Rater 2** | $\kappa$ | **Rater 1** | **Rater 2** | $\kappa$ |
| Human_0 | Human_1 | 0.949 | LLM judge_0 | LLM judge_1 | 1 | Human_0 | LLM judge_0 | 0.870 |
| Human_0 | Human_2 | 0.950 | LLM judge_0 | LLM judge_2 | 1 | Human_1 | LLM judge_0 | 0.867 |
| Human_1 | Human_2 | 0.949 | LLM judge_1 | LLM judge_2 | 1 | Human_2 | LLM judge_0 | 0.818 |

## G.3 ADDITIONAL EVALUATION SETTINGS

A system that cannot provide fully correct end-to-end results may still be helpful for end-users via assisting them in the process of data pipeline design and implementation. Motivated by the goal of assessing this type of helpfulness of systems, we conduct evaluations under two less-automated settings. In Section 4 detailing our experiments, we report these results as micro-benchmarks in Table 8.

**Pipeline Design:** This setting evaluates how many essential functions a system-generated pipeline includes. Here, we ask the system to provide an end-to-end pipeline implemented in Python that solves an end-to-end task. For evaluation, we manually curated an explicit list of key functionalities that any correct solution must implement for each task. We evaluate whether the generated pipeline code covers each functionality using the LLM evaluation method proposed in Tong & Zhang (2024). The score for a single task is computed as

$$\frac{\sum_{f \in KF(T)} \text{Judge}(f, P)}{|KF(T)|}$$

Here, $KF(T)$ denotes the set of human-annotated key functionalities for task $T$, $|KF(T)|$ is the number of those functionalities, $f$ represents a single functionality, $P$ is the pipeline the system generated under test, and Judge is a binary decision from an LLM-based evaluator indicating wether $P$ contains the key functionality $f$. The overall score across a workload/the entire benchmark is the average of the individual task scores.

**Sub-task Implementation:** This setting evaluates the system's ability to correctly implement simpler, lower-level functionalities and individual data tasks required to solve the entire challenge when explicitly prompted. We provide the system with problem statements of sub-tasks generate in Step 4 of the benchmark curation. Each sub-task corresponds to a key functionality and represents an intermediate step within the full end-to-end pipeline, operating over the gold subset of the data lake. We assess sub-task performance by comparing the system's intermediate outputs to human-annotated references, using an evaluation approach similar to the end-to-end automated method described earlier in this section.

# H    SUMMARY OF LLM USAGE

In this section, we summarize our usage of LLMs in compliance with the conference policy. We used LLMs for the following purposes

1. LLMs were used for the semi-automated generation of fine-grained annotations for the benchmark. However, contributors manually improved and verified all annotations. This is described in detail in Subsection 2.1.

2. LLMs are an integral part of the systems we evaluated. Their roles in the systems are described in detail in Subsection 2.3 and Section 3.

3. LLM-as-a-judge were used to evaluade string paraphrases and code coverage. This is described in detail in Appendix G.

4. LLMs were used to generate better documentations in our repository.

In addition to these research-level involvement of LLMs, we also used LLMs for table formatting and paraphrasing some sentences already written by authors in favor of brevity.

