# OpenReview forum: "KRAMABENCH: A Benchmark for AI Systems on Data-to-Insight Pipelines over Data Lakes"
_ICLR.cc/2026/Conference — ICLR 2026 Poster_

### Official Review · Reviewer_4eDt · 2025-10-30

**Soundness:** 4
**Presentation:** 4
**Contribution:** 4
**Rating:** 6
**Confidence:** 4

**Summary:**

This paper introduces KRAMABENCH, a large-scale benchmark designed to evaluate AI systems on end-to-end data-intensive reasoning—specifically, the ability to design and execute full data-science pipelines across heterogeneous, noisy data lakes.
The benchmark comprises 104 curated tasks drawn from 1,700 files and six domains (archaeology, astronomy, biomedical, environmental, legal, and wildfire).
Each task is derived from real-world studies and paired with reference solutions, sub-task annotations, and evaluation scripts.

The authors propose three complementary evaluation settings:

End-to-end automation – overall task correctness;

Pipeline design – coverage of key functional components;

Sub-task implementation – correctness on atomic operations.

A minimal baseline framework, DS-Guru, enables single-agent LLMs (e.g., GPT-4o, Claude-3.5, DeepSeek-R1, Llama3.3) to attempt these challenges.
The benchmark also evaluates agentic systems such as Hugging Face Smolagents, OpenAI Deep Research, and Gemini Agentic Mode.

**Strengths:**

Originality: First benchmark targeting end-to-end data-to-insight reasoning across heterogeneous data lakes.

Quality: Comprehensive curation and evaluation pipeline; transparent methodology with open-source assets.

Clarity: Well-written with detailed ablations, case studies, and ethical considerations.

Significance: Reveals key weaknesses of state-of-the-art LLM agents (e.g., data-dependent reasoning), guiding future research in automated data science.

Breadth: Covers six domains and multiple data modalities (structured, semi-structured, unstructured).

Practical utility: Enables reproducible assessment of both academic and commercial agentic frameworks.

**Weaknesses:**

Limited evaluation diversity: While the benchmark spans multiple domains, all experiments are English-only and rely on a single execution environment; multilingual or multimodal tasks would strengthen generality.

Absence of multi-agent baselines: The paper focuses on single-agent and light agentic systems; evaluating collaborative agents or planner–executor architectures would provide additional insight.

Computation cost reporting: Token-level cost is discussed but system runtime and hardware variance could be presented more rigorously.

Limited human comparison: Including a human-in-the-loop baseline would contextualize model performance relative to expert data scientists.

Statistical significance: Some percentage improvements (e.g., +1.3 %) could benefit from confidence intervals or variance analysis.

**Questions:**

1-How does KRAMABENCH handle tasks with stochastic or non-deterministic outputs (e.g., probabilistic models)?

2- Could future iterations support incremental learning or multi-agent collaboration settings?

3- Did the authors observe any correlations between model size and success on specific domains (e.g., biomedical vs. legal)?

4- Are the obscured-input experiments sufficient to rule out partial memorization—could LLMs have seen similar structure from public repositories?

5- Would integrating structured metadata (schemas, ontology hints) into DS-Guru improve pipeline design?

---

> ### Author Response · Authors · 2025-11-21
> **Response Summary**
>
> We appreciate your thorough and detailed review. We are actively addressing the opportunities and concerns you identified. As we continue refining the submission, we will share and update our results in this discussion thread during the coming author feedback phase.
> At a high level, our ongoing efforts focus on:
>
> * Evaluate two multi-agent baselines on Kramabench: Reflexion and PDT (Planner → Decomposer → Tool Executor
> * Perform fine-grained quantitative and qualitative analysis of systems' performances variance across domains
> * Engage in a small-scale user study to provide a reference human expert score on Kramabench
> * Provide more analysis for the complexity (cost and runtime) of running Kramabench.
> * Provide more experimental evidence to strengthen the statistical significance of the results shown in the paper
> * Calibrate the LLM-based evaluation scores against human annotations and provide inter-annotator agreements.
> * Extend the obscured experiment and corresponding results to include an obfuscated version of all tasks across all domains
> * Discuss more background motivation in Section 1 to give insights into the novelty and challenges of Kramabench compared to other existing benchmarks
> * Address readability and clarity concerns by Improving the quantity and quality of the visualizations and plots provided in the paper.

---

> > ### Author Response · Authors · 2025-11-21
> > **Dataset and Task Diversity**
> >
> > We recognize the limitations related to KramaBench’s datasets size, origin, and domain coverage, and we fully agree that the community would benefit from datasets spanning a wider range of modalities and languages. These limitations stem from the substantial amount of expert human effort required to guarantee high quality and fidelity to realistic data pipelines (as detailed in Section 2). The English-centric bias is similarly a consequence of the language proficiencies of the contributors involved in building the benchmark.
> > Our goal for this initial release is to stimulate broader interest in the challenges of data intensive tasks and to provide a framework for standardized evaluation. We intend to continue expanding KramaBench over time and will actively seek contributions from the community. Future workloads will be both more specialized and more expansive, incorporating larger and more varied input datasets informed by community feedback. Through our open source repository we welcome collaborations with diverse collaborators and encourage the submission of new datasets and tasks beyond English.

---

> > > ### Author Response · Authors · 2025-11-21
> > > **Benchmarking Multi-agent Systems**
> > >
> > > We thank the reviewer for the suggestion to include multi-agent baselines. We agree this is an important dimension of evaluating real-world data-to-insight agents.
> > >
> > > Following this feedback, we have implemented and begun evaluating two state-of-the-art multi-agent architectures:
> > > * Reflexion (Shinn et al., 2023) — an iterative plan–critique–revise framework.
> > > * PDT (Planner → Decomposer → Tool Executor) — a hierarchical, tool-augmented architecture aligned with recent programmatic agent systems such as DSPy (Gao et al., 2023) and the natural-language data-preparation system AutoPrep (Li et al., 2025), both of which decompose tasks into structured subtasks and execute them through tool-based Python modules.
> > >
> > > Reflexion and PDT are considered state-of-the-art because they embody the two most empirically validated advances in modern LLM agents: iterative self-correction and hierarchical, tool-grounded workflow execution. Reflexion (Shinn et al., 2023) showed across established benchmarks that adding a structured plan → act → critique → revise loop reliably improves long-horizon task performance, making it a widely adopted baseline. PDT-style architectures follow the strongest trend demonstrated in systems like DSPy (Gao et al., 2023), where agents achieve significantly higher accuracy and stability by separating global planning, explicit decomposition, and precise tool-based execution instead of relying on a single monolithic LLM. Together, these two designs represent the leading approaches for making agents more reliable, controllable, and scalable.
> > >
> > > Both architectures are implemented using smolagents, ensuring consistency with our existing setup. Preliminary Reflexion (full input mode) results are complete, and full PDT evaluation is underway. Early evidence suggests that while multi-agent coordination can improve robustness and intermediate verification, these systems still struggle with the complex, multi-step pipelines represented in KramaBench.
> > >
> > > In the revised paper, we will include full results across all six domains, along with a comparison table and discussion.
> > >
> > > |  | Archaeology | Astronomy | Biomedical | Environment | Legal  | Wildfire | **Overall** |
> > > |---|---|---|---|----|---|---|---|
> > > | smolagents Reflexion (claude-3-7-sonnet-latest) |50%| 33.33%| 44.44%| 80%| 66.67%| 47.62%| 57.69%|
> > >
> > > KramaBench is a living benchmark with a public leaderboard, and we welcome further suggestions for additional multi-agent baselines.
> > > While the author feedback phase is still ongoing, we are committed and open to perform any additional experiment and welcome any further feedback or suggestion to strengthen our submission and improve our research.

---

> > > > ### Author Response · Authors · 2025-11-21
> > > > **Detailed Cost Analysis**
> > > >
> > > > We agree that providing more insights into the complexity and cost of running the benchmark is valuable for future research.
> > > >
> > > > To this end, we implemented more fine-grained cost and runtime tracking that allows for an in-depth analysis of the execution of the benchmark:
> > > > * We explicitly differentiate between input tokens and output tokens (which may have different costs)
> > > > * We separate the tokens used for executing the benchmark using a baseline, compared to the tokens used for evaluating the benchmark results
> > > >
> > > >
> > > > Some preliminary results for our complexity analysis are as follows:
> > > >
> > > > | sut | overall | score/runtime | score/thousand input_token | score/thousand output_token |
> > > > |:-----------------------------------------|----------:|----------------:|-----------------------------:|------------------------------:|
> > > > | GPTo3 - Naive | 4.4272 | 0.0253 | 3.6242 | 2.0076 |
> > > > | GPTo3 - One Shot | 14.3006 | 0.0792 | 0.3651 | 8.0089 |
> > > > | GPTo3 - Few Shot | 26.1561 | 0.1123 | 0.3571 | 8.8960 |
> > > > | GPT4o - Naive | 1.3532 | 0.0081 | 1.1068 | 1.4968 |
> > > > | GPT4o - One Shot | 9.8278 | 0.0624 | 0.7268 | 15.6457 |
> > > > | GPT4o - Few Shot | 11.8930 | 0.0566 | 0.3718 | 9.9949 |
> > > > | Llama3_3Instruct - Naive | 1.3755 | 0.0077 | 1.1041 | 1.5074 |
> > > > | Llama3_3Instruct - One Shot | 5.9167 | 0.0313 | 0.4362 | 10.0508 |
> > > > | Llama3_3Instruct - Few Shot | 9.3734 | 0.0412 | 0.3027 | 9.9297 |
> > > > | DeepseekR1 - Naive | 3.1111 | 0.0546 | 3.0126 | 1.5262 |
> > > > | DeepseekR1 - One Shot | 2.7946 | 0.0300 | 0.0697 | 1.3719 |
> > > > | DeepseekR1 - Few Shot | 6.0351 | 0.0592 | 0.1482 | 2.9354 |
> > > >
> > > >
> > > > The most cost-effective methods in terms of input tokens across different LLMs are the naive variants (i.e., where the system is invoked with only the task and not data). This is to be expected since supplying data to the system incurs orders of magnitude more input token costs.
> > > >  All few-shot systems with different LLM backends have roughly the same accuracy per input token. Systems roughly achieve 0.35% more accuracy per thousand input tokens additively.
> > > > Looking at the ratio between the number of output tokens and accuracy to measure effectiveness of system outputs, one-shot systems are the best. The best few-shot system for accuracy per output token is GPT-4o few-shot, gaining ~10% per thousand output tokens additively.
> > > >
> > > > In terms of accuracy to runtime ratio, GPT-o3 performs the best. Across different LLM backends, few-shot and one-shot do better than the naive (no data context) variant.
> > > >
> > > > As we are running our experiments, we will keep updating this table and report the complete numbers in a revised version of the paper.

---

> > > > > ### Author Response · Authors · 2025-11-21
> > > > > **Human Baseline**
> > > > >
> > > > > We agree with the reviewer that including a human expert comparison is valuable for contextualizing LLM performance. To address this, we **have already designed and initiated a human data-science expert study** for KramaBench. The study recruits 10 domain-knowledgeable data scientists and instructs them to solve a subset of tasks using the same data directory structure and constraints as our LLM agents. Each expert produces:
> > > > > * A complete, reproducible end-to-end solution in a Jupyter notebook.
> > > > > * A full record of their active time per task, with a breakdown across data exploration, pipeline design, coding, and debugging.
> > > > > *  Draft-stage notes and final clean solutions, allowing direct comparison to LLM workflows.
> > > > >
> > > > > We provide experts with detailed instructions on repository setup, task assignment, time tracking, and reproducibility, and we prevent direct LLM usage to ensure the baseline reflects genuine human reasoning and coding effort.
> > > > >
> > > > > We are currently collecting solutions and time measurements across multiple domains. In the revised version, we will report:
> > > > > 1. Human solution accuracy
> > > > > 2. Active time per task compared to LLM system runtimes
> > > > > 3. Qualitative observations highlighting human vs. LLM strengths (e.g., data discovery, handling messy schema, complex joins)
> > > > >
> > > > > This baseline will provide a clear picture of how far current agentic systems are from expert-level data-science performance, and we plan to expand it as KramaBench evolves.

---

> > > > > > ### Author Response · Authors · 2025-11-21
> > > > > > **Statistical Significance of Results**
> > > > > >
> > > > > > We are committed to providing statistically significant results and measurements. For that reason, we are repeating the whole benchmark three times, and in a revised version of the paper, we will include confidence bounds for every metric with average and standard deviation derived from the three runs. Being mindful about the extensive costs of running the whole benchmark pipeline under the different scenarios, and considering our stable results across runs, we believe 3 repetitions to be a sufficient number.
> > > > > > We will update our response with the results of our experiments during the upcoming author feedback weeks.

---

> ### Author Response · Authors · 2025-11-21
> **Answering Further Questions**
>
> ### Non-deterministic Outputs
> Kramabench includes only a few tasks that require stochastic models with simple data-driven models (e.g., logistic regression and polynomial interpolation). For these tasks, we measure their accuracy using an approximate measure (i.e., relative absolute error) so we do not require exact answers for a system to score well. Given the simplicity of the numerical models required, and the fixed input data points, we do not expect huge variance for correct  outputs.
>
> ### Incremental learning
> Yes, the design of Kramabench allows the implementation of further baselines with more sophisticated interactions between agents. We are committed into benchmarking state of the art systems as well as making our codebase open and accessible to external contributors. We plan to keep an up-to-date version of our results on an online leaderboard.
>
> ### Limitation of obscured experiments
> We acknowledge the limitations of the obscured experiment setup, as the structure and general pipeline design of correct solutions is not affected by the obfuscation process. However, since the tasks are inspired by real-world existing data science pipelines across domain specific studies, we believe that any knowledge absorbed by LLMs from public sources is likely beneficial in solving these tasks. To further generalize our results, we are obscuring the full input datasets (as mentioned in response to other reviewers). We are open to further suggestions to mitigate the effect of memoization in our experiments.
>
> ### Correlation between model sizes and performance
>
> Based on the benchmark results, there is no strict linear correlation between model size and performance success. While the most capable model (GPT-o3) generally leads, smaller domain-specific models occasionally outperform larger generalist models. For instance, in the "few-shot" setting, the smaller Qwen2.5-Coder-32B achieved a higher overall score (9.98%) than the significantly larger DeepSeek-R1 (6.34%) and even GPT-4o (8.28%). This discrepancy is visible across domains. In the Legal domain, Qwen (9.06%) and DeepSeek-R1 (8.39%) are on par or out-perform models like GPT-o3 (13.89%), GPT-4o (2.80%) or Claude-3.5 (7.01%)
> However, in the few-shot archeology domain, GPT-o3 (25%), GPT-4o (16.67%), and Claude-3.5 (16.67%) outperformed models like Qwen2.5Coder (8.33%) and DeepSeek-R1 (8.33%).
> This indicates that success on Kramabench relies more on specific capabilities which are more used in varying domains (see the above response for which domains use varying skills).
>
> ### Integrate structured metadata into DS-GURU
> As our experiments on DSGURU with OPS show, by feeding more information about data structure and schemas in the prompts, systems are able to reach better results. Considering the results of Table 5, the overall scores of DS-GURU in the one and few-shot variants consistently outperform the no-context variants by ~10 percentage points. We expect that future systems which include more sophisticated data reasoning agents and structured information about input data, e.g.,  through the use of ontologies, will reach even higher scores.

---

### Official Review · Reviewer_WbPU · 2025-11-01

**Soundness:** 3
**Presentation:** 3
**Contribution:** 2
**Rating:** 4
**Confidence:** 3

**Summary:**

1. This paper introduces KRAMABENCH, a benchmark designed to evaluate AI systems’ ability to solve end-to-end data science pipelines on real-world data lakes (heterogeneous, unclean, multi-source data). It includes 104 manually curated tasks across 6 domains (e.g., archaeology, biomedical research), 1700 files, and 24 data sources, paired with expert reference solutions and a 3-tier evaluation framework (end-to-end automation, pipeline design, sub-task implementation). The authors also propose DS-Guru, a lightweight single-agent framework for baseline testing, and evaluate 8 LLMs + 4 agentic systems (e.g., smolagents DR, OpenAI Deep Research).

**Strengths:**

1. Fills a critical gap in existing benchmarks (e.g., DS-1000, ARCADE) by focusing on end-to-end data lake processing rather than isolated tasks (code generation, text-to-SQL). Unlike prior work, it emphasizes real-world complexity (noisy data, multi-file integration, domain-specific knowledge) and requires systems to orchestrate all pipeline stages (discovery, cleaning, analysis).
2. Rigorous task curation via a 4-step validation process (curation → cross-contributor verification → key functionality identification → sub-task curation) ensures high-quality, reproducible reference solutions.
3. Extensive experiments across 8 LLMs, 4 agentic systems, and 3 input modes (Full, Oracle, Trimmed) provide statistically meaningful results. Ablation studies (e.g., retrieval mechanism, sample size, iterations) further validate conclusions about performance drivers.

**Weaknesses:**

1. The paper focuses on single-agent systems (DS-Guru, smolagents DR) but barely explores multi-agent architectures, which are increasingly proposed for complex data tasks (e.g., dividing pipeline stages across specialized agents). This is a missed opportunity, as multi-agent systems may mitigate single-agent limitations (e.g., context window constraints, heterogeneous skill requirements).
2. The results show large performance gaps across domains (e.g., smolagents DR achieves 60% accuracy in Environment but 16.67% in Astronomy), but the paper provides minimal analysis of why (e.g., is Astronomy’s larger file size, more complex data formats, or domain unfamiliarity the cause?). Without this, readers cannot generalize findings to other domains.

**Questions:**

Beyond Weaknesses. The following is the suggestions for the authors.
1. Incorporate Human-in-the-Loop Evaluation: Since real-world data science often involves human-AI collaboration, add a fourth evaluation tier (e.g., human oversight of pipeline revisions) to measure how systems assist human experts. This would increase the benchmark’s relevance for industry use cases.
2. Scalability and Cost Analysis: Extend experiments to larger data lakes (e.g., 10k+ files) to test scalability of retrieval mechanisms (OPS vs. agentic retrieval). Add cost-per-accuracy metrics (e.g., dollars/task, energy consumption) to guide practical adoption.

---

> ### Author Response · Authors · 2025-11-21
> **Response Summary**
>
> We appreciate your comprehensive and detailed review. We are currently taking steps to address the opportunities and concerns raised in your comments. As we are actively working on the submission, we will update the results throughout the upcoming weeks in this discussion thread.
> At a glance, to address all reviewer concerns, we are actively working to:
>
> * Evaluate two multi-agent baselines on Kramabench: Reflexion and PDT (Planner → Decomposer → Tool Executor
> * Perform fine-grained quantitative and qualitative analysis of systems' performances variance across domains
> * Engage in a small-scale user study to provide a reference human expert score on Kramabench
> * Provide more analysis for the complexity (cost and runtime) of running Kramabench.
> * Calibrate the LLM-based evaluation scores against human annotations and provide inter-annotator agreements.
> * Extend the obscured experiment and corresponding results to include an obfuscated version of all tasks across all domains
> * Discuss more background motivation in Section 1 to give insights into the novelty and challenges of Kramabench compared to other existing benchmarks
> * Address readability and clarity concerns by Improving the quantity and quality of the visualizations and plots provided in the paper

---

> > ### Author Response · Authors · 2025-11-21
> > **Benchmarking Multi-agent Systems**
> >
> > We thank the reviewer for the suggestion to include multi-agent baselines. We agree this is an important dimension of evaluating real-world data-to-insight agents.
> >
> > Following this feedback, we have implemented and begun evaluating two state-of-the-art multi-agent architectures:
> > * Reflexion (Shinn et al., 2023) — an iterative plan–critique–revise framework.
> > * PDT (Planner → Decomposer → Tool Executor) — a hierarchical, tool-augmented architecture aligned with recent programmatic agent systems such as DSPy (Gao et al., 2023) and the natural-language data-preparation system AutoPrep (Li et al., 2025), both of which decompose tasks into structured subtasks and execute them through tool-based Python modules.
> >
> > Reflexion and PDT are considered state-of-the-art because they embody the two most empirically validated advances in modern LLM agents: iterative self-correction and hierarchical, tool-grounded workflow execution. Reflexion (Shinn et al., 2023) showed across established benchmarks that adding a structured plan → act → critique → revise loop reliably improves long-horizon task performance, making it a widely adopted baseline. PDT-style architectures follow the strongest trend demonstrated in systems like DSPy (Gao et al., 2023), where agents achieve significantly higher accuracy and stability by separating global planning, explicit decomposition, and precise tool-based execution instead of relying on a single monolithic LLM. Together, these two designs represent the leading approaches for making agents more reliable, controllable, and scalable.
> >
> > Both architectures are implemented using smolagents, ensuring consistency with our existing setup. Preliminary Reflexion (full input mode) results are complete, and full PDT evaluation is underway. Early evidence suggests that while multi-agent coordination can improve robustness and intermediate verification, these systems still struggle with the complex, multi-step pipelines represented in KramaBench.
> >
> > In the revised paper, we will include full results across all six domains, along with a comparison table and discussion.
> >
> > |  | Archaeology | Astronomy | Biomedical | Environment | Legal  | Wildfire | **Overall** |
> > |---|---|---|---|----|---|---|---|
> > | smolagents Reflexion (claude-3-7-sonnet-latest) |50%| 33.33%| 44.44%| 80%| 66.67%| 47.62%| 57.69%|
> >
> > KramaBench is a living benchmark with a public leaderboard, and we welcome further suggestions for additional multi-agent baselines.
> > While the author feedback phase is still ongoing, we are committed and open to perform any additional experiment and welcome any further feedback or suggestion to strengthen our submission and improve our research.

---

> > > ### Author Response · Authors · 2025-11-21
> > > **Analysis of Performance Gaps**
> > >
> > > We appreciate the feedback that the benchmark results presented in the paper would be more interpretable with a deeper analysis on the performance gap between different domains.
> > > We will update the discussion section of the paper to include a quantitative analysis of the per-domain difficulty of tasks;  we will also post this to Open Review before the final deadline.
> > >
> > > To keep this response concise, we provide a high-level summary of our findings for the system with the highest result variance (smolagents DR with Claude 3.7):
> > >
> > > ### Domain-specific performance patterns
> > >
> > > * **Archaeology (33.33%)** : In this domain, the system correctly solves questions answerable from a single table. However, errors occur for tasks requiring joining tables found in different files, because it treats multiple files as raw text instead of loading them as tables.
> > > * **Astronomy (16.67%)** : Astronomy tasks have the lowest average performance. In this domain, a large portion of the required input data is found in proprietary scientific formats (e.g., FORTRAN-style dat files). We observed that the agent struggles whenever it needed to load data from these files, e.g., SP3 orbit files or satellite products.
> > > * **Biomedical (44.44%)** : When working with biomedical data, the agent is reliable for shallow operations on a single sheet but fails to navigate large, multi-sheet workbooks and join data across sheets. Cross-sheet joins, especially between clinical and phosphoproteomics data, are problematic, and errors arise in correlation statistics due to sign miscalculations.
> > > * **Environment (60.00%)** : In the environmental domain, the system performs well on the relatively tasks involving clean CSV data, such as filtering, counting, and averaging. Unlike other tasks which struggle from data retrieval or understanding issues, the main issues arise from implementation/arithmetic mistakes, such as incorrect aggregation scopes or rounding errors, which explains the higher score.
> > > * **Legal (63.33%)**: In these tasks, the agent handles the straightforward pipelines well but struggles with loading data from messy files, i.e., that contain multi-row headers, partial subtotals, and metadata rows. Amongst the common errors that stem from these shortcomings, one example is that sum, means, and aggregations are only partial due to incorrect loading.
> > > * **Wildfire (52.38%)** : Within wildfire-related tasks, the system faces challenges with geospatial data and temporal/statistical reasoning. It struggles with GeoPackage layers and spatial joins, as well as with rolling-window aggregations for weather. However, text lookups and simple value comparisons work relatively well.

---

> ### Author Response · Authors · 2025-11-21
> **Human Baseline**
>
> We agree with the reviewer that including a human expert comparison is valuable for contextualizing LLM performance. To address this, we **have already designed and initiated a human data-science study** for KramaBench. The study recruits 9 participants and instructs them to solve a subset of tasks using the same data directory structure and constraints as our LLM agents. Each participant produces:
>
> * A complete, reproducible end-to-end solution in a Jupyter notebook.
> * A full record of their active time per task, with a breakdown across data exploration, pipeline design, coding, and debugging.
> * Draft-stage notes and final clean solutions, allowing direct comparison to LLM workflows.
>
> We provide participants with detailed instructions on repository setup, task assignment, time tracking, and reproducibility, and we prevent direct LLM usage to ensure the baseline reflects genuine human reasoning and coding effort.
>
> We are currently collecting solutions and time measurements across multiple domains. In the revised version, we will report:
>
> 1. Human solution accuracy
> 2. Active time per task compared to LLM system runtimes
> 3. Qualitative observations highlighting human vs. LLM strengths (e.g., data discovery, handling messy schema, complex joins)
>
> This baseline will provide a clear picture of how far current agentic systems are from human data-science performance, and we plan to expand it as KramaBench evolves.

---

> ### Author Response · Authors · 2025-11-21
> **Detailed Cost Analysis**
>
> We agree that providing more insights into the complexity and cost of running the benchmark is valuable for future research.
>
> To this end, we implemented more fine-grained cost and runtime tracking that allows for an in-depth analysis of the execution of the benchmark:
>
> * We explicitly differentiate between input tokens and output tokens (which may have different costs)
> * We separate the tokens used for executing the benchmark using a baseline, compared to the tokens used for evaluating the benchmark results
>
>
>
> Some preliminary results for our complexity analysis are as follows:
>
> | sut | overall | score/runtime | score/thousand input_token | score/thousand output_token |
> |:-----------------------------------------|----------:|----------------:|-----------------------------:|------------------------------:|
> | GPTo3 - Naive | 4.4272 | 0.0253 | 3.6242 | 2.0076 |
> | GPTo3 - One Shot | 14.3006 | 0.0792 | 0.3651 | 8.0089 |
> | GPTo3 - Few Shot | 26.1561 | 0.1123 | 0.3571 | 8.8960 |
> | GPT4o - Naive | 1.3532 | 0.0081 | 1.1068 | 1.4968 |
> | GPT4o - One Shot | 9.8278 | 0.0624 | 0.7268 | 15.6457 |
> | GPT4o - Few Shot | 11.8930 | 0.0566 | 0.3718 | 9.9949 |
> | Llama3_3Instruct - Naive | 1.3755 | 0.0077 | 1.1041 | 1.5074 |
> | Llama3_3Instruct - One Shot | 5.9167 | 0.0313 | 0.4362 | 10.0508 |
> | Llama3_3Instruct - Few Shot | 9.3734 | 0.0412 | 0.3027 | 9.9297 |
> | DeepseekR1 - Naive | 3.1111 | 0.0546 | 3.0126 | 1.5262 |
> | DeepseekR1 - One Shot | 2.7946 | 0.0300 | 0.0697 | 1.3719 |
> | DeepseekR1 - Few Shot | 6.0351 | 0.0592 | 0.1482 | 2.9354 |
>
> The most cost-effective methods in terms of input tokens across different LLMs are the naive variants (i.e., where the system is invoked with only the task and not data). This is to be expected since supplying data to the system incurs orders of magnitude more input token costs.
>  All few-shot systems with different LLM backends have roughly the same accuracy per input token. Systems roughly achieve 0.35% more accuracy per thousand input tokens additively.
> Looking at the ratio between the number of output tokens and accuracy to measure effectiveness of system outputs, one-shot systems are the best. The best few-shot system for accuracy per output token is GPT-4o few-shot, gaining \~10% per thousand output tokens additively.
>
> In terms of accuracy to runtime ratio, GPT-o3 performs the best. Across different LLM backends, few-shot and one-shot do better than the naive (no data context) variant.
>
> As we are running our experiments, we will keep updating this table and report the complete numbers in a revised version of the paper.

---

### Official Review · Reviewer_fJHC · 2025-11-02

**Soundness:** 2
**Presentation:** 1
**Contribution:** 2
**Rating:** 4
**Confidence:** 4

**Summary:**

This paper introduce KramaBench, a benchmark to evaluate LLM-based systems on data science tasks. KramaBench includes 104 mannually curated tasks from 1700 real-world files across 24 sources in 6 domains (archaeology, astronomy, biomedical research, environmental science, legal discovery, and wildfire prevention). The authors then provide a comprehensive evaluation framwork with several settings (1) end-to-end automation, (2) pipeline design, and (3) individual task implementation. With experiments on 8 LLMs using DS-Guru reference framework and several agentic systems, the authors find that current systems struggle with end-to-end pipeline generation, even with perfect retrieval.

**Strengths:**

- **Comprehensive Evaluation**: The multi-level evaluation framework (end-to-end, pipeline design, sub-task implementation) provides valuable insights into where systems fail.
- **Rigorous Curation Process**: The 4-step validation process involving multiple contributors, cross-validation, and manual verification of reference solutions demonstrates strong quality control. Grounding tasks in published studies ensures real-world relevance and avoids artificial task design.
- **Diverse Tasks**: The benchmark spans multiple domains with varying data formats, file counts, and difficulty levels, providing good coverage of real-world scenarios.

**Weaknesses:**

- **Unclear Motivation and Weak Problem Positioning**: The paper lacks a compelling motivation section explaining why this benchmark is needed now and what specific real-world problems it addresses that existing benchmarks cannot. The introduction jumps directly into the solution without establishing the problem's urgency or providing concrete use cases where current benchmarks fail.
- **Poor Figure Quality**: Figure 1 is a low-resolution raster image with blurry, barely readable text, which is unacceptable for a top-tier venue. The figure should be vector graphics (SVG/PDF) with crisp, readable text. This significantly impacts the paper's professionalism and readability.
- **Severely Insufficient Visual Explanations**: The paper has only 2 figures across 9 pages of main content, making it difficult to understand key components. Critical missing visualizations include (1) DS-Guru architecture diagram showing the three variants and data flow, (2) benchmark curation pipeline flowchart for the 4-step process, (3) task structure schema showing relationships between end-to-end tasks, key functionalities, and sub-tasks, (4) evaluation framework diagram.

**Questions:**

- What is the conceptual or methodological novelty beyond creating a benchmark with more realistic data science tasks?
- Can you provide validation of the LLM judge against human evaluators, including agreement rates and calibration analysis?
- Can you provide detailed computational cost analysis (tokens, time, dollars) for running each system on the full benchmark?

---

> ### Author Response · Authors · 2025-11-21
> **Response Summary**
>
> Thank you for the thorough and thoughtful review. We are actively working to address the opportunities and concerns raised in your comments, and will post updated results in this OpenReview thread throughout the author feedback phase in the next weeks.
> In summary, the major steps we are taking to mitigate the concerns expressed in all the reviews we received are:
>
> * Discuss more background motivation in Section 1 to give insights into the novelty and challenges of Kramabench compared to other existing benchmarks.
> * Improve the quantity and quality of the visualizations and plots provided in the paper to address readability and clarity concerns. We updated the submission PDF implementing all recommended suggestions.
> * Calibrate the LLM-based evaluation scores against human annotations and provide inter-annotator agreements.
> * Provide more analysis for the complexity (cost and runtime) of running Kramabench.
> * Extend the obscured experiment and corresponding results to include an obfuscated version of all tasks across all domains
> * Perform fine-grained quantitative and qualitative analysis of systems' performances variance across domains
> * Evaluate two multi-agent baselines on Kramabench: Reflexion and PDT (Planner → Decomposer → Tool Executor
> * Engage in a small-scale user study to provide a reference human expert score on Kramabench

---

> > ### Author Response · Authors · 2025-11-21
> > **Positioning and motivation**
> >
> > We acknowledge that the current introduction could better articulate the motivation and importance of KramaBench. To fit within space constraints, we plan to add an upfront paragraph in Section 1 to discuss the shortcomings of current benchmarks and the main novelty of KramaBench - discussing the content of Table 1.
> > In particular, we will emphasize that KramaBench is the only end-to-end benchmark designed for data-science workflows where systems must operate over large, uncurated data lakes, across a wide variety of domain-specific data pipelines. This unique combination of challenges is not fully captured by any of the established benchmarks seen in Table 1, e.g., ScienceAgentBench, DataSciBench, or DSBench, which directly provide systems with the correct input data and do not test for pipeline design and implementation.

---

> > > ### Author Response · Authors · 2025-11-21
> > > **Presentation Quality**
> > >
> > > We understand that clear, high-quality visuals are essential to improve the readability of the paper and better convey our fundamental contributions. We acknowledge that Figure 1’s current format is inadequate, and in the updated PDF of the submission we revised the figure to replace it with a high-resolution and polished version in pdf format.
> > >
> > > More broadly, we recognize the need for additional visual explanations. We appreciate the concrete recommendations provided and have incorporated all of them in the revised version of our paper – which we uploaded as a PDF.
> > >
> > >  Specifically, we added:
> > > * A clear architecture diagram illustrating the three DS-Guru variants and their data flows;
> > > * A flowchart of the full benchmark curation pipeline;
> > > * A task-structure schema showing how end-to-end tasks, key functionalities, and sub-tasks relate; and
> > > * An expanded evaluation framework diagram.
> > >
> > > Improving the presentation of the paper is a priority for us, and we welcome any further feedback.

---

> > > > ### Author Response · Authors · 2025-11-21
> > > > **Human Agreement and LLM-Evaluation Calibration**
> > > >
> > > > We agree that calibrating the LLM-based judge against human evaluators provides important information to ensure the reliability of our evaluation methodology. Toward this end, we conducted a dedicated manual annotation study in which three human annotators independently labeled the model outputs that were also evaluated by our LLM judge. We performed this study across all workloads, and for 12 system configurations (4 backend with the three naive, one shot and few shot variants) to ensure broad coverage. We repeated the LLM judge evaluation 3 times. We exclude tasks for which the output of a systems was an error message caught through our benchmark pipeline.
> > > > We report the Cohen's Kappa values for inter-human agreement, human-LLM agreement and inter-LLM calibration. The possible values range from -1 (complete misalignment) to 1 (complete alignment):
> > > >
> > > > | Rater 1     | Rater 2     |   Kappa |
> > > > |:--------------|:--------------|---------:|
> > > > | Human_0 | Human_1 | 0.949416 |
> > > > | Human_0 | Human_2 | 0.950273 |
> > > > | Human_1 | Human_2 | 0.949416 |
> > > >
> > > > | Rater 1     | Rater 2     |   Kappa |
> > > > |:------------|:------------|--------:|
> > > > |  LLM judge_0 | LLM judge_1 |       1 |
> > > > |  LLM judge_0 | LLM judge_2 |       1 |
> > > > |  LLM judge_1 | LLM judge_2 |       1 |
> > > >
> > > > | Rater 1     | Rater 2     |   Kappa |
> > > > |:--------------|:------------|---------:|
> > > > | Human_0 | LLM judge_0 | 0.870185 |
> > > > | Human_1 | LLM judge_0 | 0.867076 |
> > > > | Human_2 | LLM judge_0 | 0.81826  |
> > > >
> > > >
> > > > To calibrate our results, inter-human agreement is very high, with scores ~0.95. Similarly, the results of our analysis show that the LLM judge produces highly consistent scores between runs with perfect agreement. The human-LLM agreement is very high (positive with a minimum of 0.81 and a max of 0.87), although not perfect, with a slight misalignment between LLM and human annotators.
> > > > We will include a discussion of these results, the full agreement statistics, and all methodological details in the paper, as well as the code to reproduce our analysis in the repository.

---

> > > > > ### Author Response · Authors · 2025-11-21
> > > > > **Detailed Cost Analysis**
> > > > >
> > > > > We agree that providing more insights into the complexity and cost of running the benchmark is valuable for future research.
> > > > >
> > > > > To this end, we implemented more fine-grained cost and runtime tracking that allows for an in-depth analysis of the execution of the benchmark:
> > > > > We explicitly differentiate between input tokens and output tokens (which may have different costs)
> > > > > We separate the tokens used for executing the benchmark using a baseline, compared to the tokens used for evaluating the benchmark results
> > > > >
> > > > >
> > > > > Some preliminary results for our complexity analysis are as follows:
> > > > >
> > > > > | sut | overall | score/runtime | score/1k input_token | score/1k output_token |
> > > > > |:-----------------------------------------|----------:|----------------:|-----------------------------:|------------------------------:|
> > > > > | GPTo3 - Naive | 4.4272 | 0.0253 | 3.6242 | 2.0076 |
> > > > > | GPTo3 - One Shot | 14.3006 | 0.0792 | 0.3651 | 8.0089 |
> > > > > | GPTo3 - Few Shot | 26.1561 | 0.1123 | 0.3571 | 8.8960 |
> > > > > | GPT4o - Naive | 1.3532 | 0.0081 | 1.1068 | 1.4968 |
> > > > > | GPT4o - One Shot | 9.8278 | 0.0624 | 0.7268 | 15.6457 |
> > > > > | GPT4o - Few Shot | 11.8930 | 0.0566 | 0.3718 | 9.9949 |
> > > > > | Llama3_3Instruct - Naive | 1.3755 | 0.0077 | 1.1041 | 1.5074 |
> > > > > | Llama3_3Instruct - One Shot | 5.9167 | 0.0313 | 0.4362 | 10.0508 |
> > > > > | Llama3_3Instruct - Few Shot | 9.3734 | 0.0412 | 0.3027 | 9.9297 |
> > > > > | DeepseekR1 - Naive | 3.1111 | 0.0546 | 3.0126 | 1.5262 |
> > > > > | DeepseekR1 - One Shot | 2.7946 | 0.0300 | 0.0697 | 1.3719 |
> > > > > | DeepseekR1 - Few Shot | 6.0351 | 0.0592 | 0.1482 | 2.9354 |
> > > > >
> > > > >
> > > > > The most cost-effective methods in terms of input tokens across different LLMs are the naive variants (i.e., where the system is invoked with only the task and not data). This is to be expected since supplying data to the system incurs orders of magnitude more input token costs.
> > > > >  All few-shot systems with different LLM backends have roughly the same accuracy per input token. Systems roughly achieve 0.35% more accuracy per thousand input tokens additively.
> > > > > Looking at the ratio between the number of output tokens and accuracy to measure effectiveness of system outputs, one-shot systems are the best. The best few-shot system for accuracy per output token is GPT-4o few-shot, gaining ~10% per thousand output tokens additively.
> > > > >
> > > > > In terms of accuracy to runtime ratio, GPT-o3 performs the best. Across different LLM backends, few-shot and one-shot do better than the naive (no data context) variant.
> > > > >
> > > > > As we are running our experiments, we will keep updating this table and report the complete numbers in a revised version of the paper.

---

> > > > > > ### Comment · Reviewer_fJHC · 2025-11-26
> > > > > >
> > > > > > Thanks for the comprehensive response from the author, most of my doubts have been resolved, and I will raise the score to 6.

---

### Official Review · Reviewer_9FHL · 2025-11-12

**Soundness:** 3
**Presentation:** 3
**Contribution:** 2
**Rating:** 6
**Confidence:** 3

**Summary:**

### Summary
This paper proposes KRAMABENCH, a comprehensive benchmark for evaluating AI systems’ end-to-end capabilities on data-intensive tasks. Targeting the gap of existing benchmarks that focus on isolated steps rather than full data science pipelines, KRAMABENCH comprises 104 manually curated tasks across 6 domains (archaeology, astronomy, biomedical research, etc.), covering 1700 files from 24 sources. The benchmark designs three evaluation settings (end-to-end automation, pipeline design, sub-task implementation) and evaluates 8 LLMs alongside multiple agentic systems (e.g., smolagents DR, OpenAI DR). Key findings include: current systems struggle with end-to-end pipeline generation (top accuracy 50%), agentic control flows significantly outperform structured loops, and fine-grained data-dependent reasoning and holistic data lake understanding are major bottlenecks. The paper also provides detailed ablation studies, failure analyses, and full reproducible artifacts, offering valuable insights for future research on automated data science systems.

### Strengths
1. **Gap-Filling Innovation**: Addresses a critical limitation of existing benchmarks by focusing on end-to-end data science pipelines (from data discovery to insight generation) rather than isolated tasks (e.g., code generation, text-to-SQL), aligning with real-world data science workflows.
2. **Comprehensive Design**: Covers diverse domains, heterogeneous data types (structured/semi-structured/unstructured), and multi-faceted evaluation dimensions, ensuring the benchmark’s generality and robustness.
3. **Rigorous Experimentation**: Evaluates a wide range of LLMs and agentic systems, conducts in-depth ablation studies (retrieval mechanisms, data leakage, iteration counts) and failure analyses, providing conclusive evidence for key claims.
4. **High Reproducibility**: Publicly releases all code, data, workloads, and evaluation scripts, adhering to top conference standards for reproducibility and facilitating follow-up research.





### Weaknesses
1. **Limited Generalizability of Data Leakage Evaluation**: The data leakage assessment only covers 20% of tasks (focused on legal and wildfire domains) with synthetic identifiers/numerics. This narrow scope may not fully reflect how systems rely on memorized data across different domains or task types.

2. **Insufficient Analysis of Domain-Specific Performance Gaps**: The paper reports significant performance variations across domains (e.g., wildfire: ~50% accuracy vs. astronomy: ~16% for smolagents DR), but lacks in-depth analysis of why certain domains are more challenging (e.g., data complexity, domain knowledge requirements).

3. **Lack of Multi-Agent Evaluation**: Current evaluations focus on single-agent systems, but multi-agent collaboration is a promising direction for complex data-intensive tasks—omitting this dimension limits the benchmark’s coverage of state-of-the-art approaches.


### Questions
1. Given that the data leakage evaluation only covers 20% of tasks and is concentrated in the legal and wildfire domains, how do you ensure that the results obtained using synthetic data can generalize to other domains and task types to accurately reflect the characteristics of systems' dependence on memorized data?(Related to Weakness 1)

2. Regarding the significant performance gap of smolagents DR between the wildfire domain (~50% accuracy) and the astronomy domain (~16% accuracy), do you plan to supplement in-depth analysis from dimensions such as data complexity and domain knowledge requirements to explain the root causes of difficulty in different domains?(Related to Weakness 2)

3. Multi-agent collaboration has become a promising cutting-edge direction for solving complex data-intensive tasks. Why did you not incorporate this dimension in the evaluation, and does this benchmark have plans to supplement multi-agent system testing in the future to cover the latest technical approaches?(Related to Weakness 3)

**Strengths:**

See Summary

**Weaknesses:**

See Summary

**Questions:**

See Summary

---

> ### Author Response · Authors · 2025-11-21
> **Response Summary**
>
> Thank you for the detailed and insightful review. We are working to address the opportunities and concerns outlined in your comments and will update our responses here over the next two weeks as we have updates. To summarize these are the major steps we are taking to mitigate the concerns expressed in all the reviews received:
>  * Extend the obscured experiment and corresponding results to include an obfuscated version of all tasks across all domains
> * Perform fine-grained quantitative and qualitative analysis of systems' performances variance across domains
> * Evaluate two multi-agent baselines on Kramabench: Reflexion and PDT (Planner → Decomposer → Tool Executor
> * Calibrate the LLM-based evaluation scores against human annotations and provide inter-annotator agreements.
> * Address readability and clarity concerns by Improving the quantity and quality of the visualizations and plots provided in the paper.
> * Provide more analysis for the complexity (cost and runtime) of running Kramabench.
> * Engage in a small-scale user study to provide a reference human expert score on Kramabench

---

> > ### Author Response · Authors · 2025-11-21
> > **Generalizability of Evaluation**
> >
> > We agree that the claims about the generalizability of KramaBench would be stronger with a more complete obfuscation across domains and tasks. For this purpose, we are working to obfuscate all workload tasks across all domains - and will update our experimental analysis with numbers obtained on fully obfuscated datasets – which we intend to keep private as an unseen test set.
> > Here are the preliminary measurements for the fully obfuscated results of Kramabench on four out of the six domains:
> >
> >
> >
> >
> >
> >
> >
> > | System                                     |   astronomy-obscured |   environment-obscured |   legal-obscured |   wildfire-obscured |   overall |   runtime |
> > |:-----------------------------------------|---------------------:|-----------------------:|-----------------:|--------------------:|----------:|----------:|
> > | GPTo3- Naive              |                 3.17 |                   1.23 |             5.02 |                0    |      2.13 |   2887.51 |
> > | GPT4o- Naive              |                 2.05 |                   1.01 |             5.02 |                0    |      1.82 |   2529.7  |
> > | Llama3_3Instruct- Naive   |                 1.93 |                   1.01 |             4.63 |                0    |      1.72 |   1033.55 |
> > | DeepseekR1- Naive         |                 3.17 |                   2.32 |             7.88 |                0    |      3.14 |   5954.48 |
> > | GPTo3- one shot            |                 3.17 |                  12.32 |             6.99 |                7.36 |      8.43 |   3398.87 |
> > | GPT4o- one shot            |                 2.03 |                   2.32 |             9.41 |               10    |      5.22 |   1490.58 |
> > | Llama3_3Instruct- one shot |                 1.91 |                   2.32 |             6.65 |               10    |      4.64 |   6704.69 |
> > | DeepseekR1- one shot       |                 3.17 |                   2.32 |            11.06 |                0    |      3.77 |   6616.46 |
> > | GPTo3- few shot            |                 1.21 |                  26.8  |             6.58 |               20.83 |     16.45 |   7073.53 |
> > | GPT4o- few shot            |                 3.31 |                   2.32 |            10.4  |               10    |      5.67 |   2847.16 |
> > | Llama3_3Instruct- few shot |                 1.91 |                   2.32 |             9.03 |                0    |      3.12 |   7270.47 |
> > | DeepseekR1- few shot       |                 3.17 |                   2.32 |             5.01 |                0    |      2.56 |   6474.5  |

---

> > > ### Author Response · Authors · 2025-11-21
> > > **Analysis of Performance Gaps**
> > >
> > > We appreciate the feedback that the benchmark results presented in the paper would be more interpretable with a deeper analysis on the performance gap between different domains.
> > > We will update the discussion section of the paper to include a quantitative analysis of the per-domain difficulty of tasks;  we will also post this to Open Review before the final deadline.
> > >
> > > To keep this response concise, we provide a high-level summary of our findings for the system with the highest result variance (smolagents DR with Claude 3.7):
> > >
> > > ### Domain-specific performance patterns
> > >
> > > * **Archaeology (33.33%)** : In this domain, the system correctly solves questions answerable from a single table. However, errors occur for tasks requiring joining tables found in different files, because it treats multiple files as raw text instead of loading them as tables.
> > > * **Astronomy (16.67%)** : Astronomy tasks have the lowest average performance. In this domain, a large portion of the required input data is found in proprietary scientific formats (e.g., FORTRAN-style dat files). We observed that the agent struggles whenever it needed to load data from these files, e.g., SP3 orbit files or satellite products.
> > > * **Biomedical (44.44%)** : When working with biomedical data, the agent is reliable for shallow operations on a single sheet but fails to navigate large, multi-sheet workbooks and join data across sheets. Cross-sheet joins, especially between clinical and phosphoproteomics data, are problematic, and errors arise in correlation statistics due to sign miscalculations.
> > > * **Environment (60.00%)** : In the environmental domain, the system performs well on the relatively tasks involving clean CSV data, such as filtering, counting, and averaging. Unlike other tasks which struggle from data retrieval or understanding issues, the main issues arise from implementation/arithmetic mistakes, such as incorrect aggregation scopes or rounding errors, which explains the higher score.
> > > * **Legal (63.33%)**: In these tasks, the agent handles the straightforward pipelines well but struggles with loading data from messy files, i.e., that contain multi-row headers, partial subtotals, and metadata rows. Amongst the common errors that stem from these shortcomings, one example is that sum, means, and aggregations are only partial due to incorrect loading.
> > > * **Wildfire (52.38%)** : Within wildfire-related tasks, the system faces challenges with geospatial data and temporal/statistical reasoning. It struggles with GeoPackage layers and spatial joins, as well as with rolling-window aggregations for weather. However, text lookups and simple value comparisons work relatively well.

---

> > > > ### Author Response · Authors · 2025-11-21
> > > > **Benchmarking  Multi-agent Systems**
> > > >
> > > > We thank the reviewer for the suggestion to include multi-agent baselines. We agree this is an important dimension of evaluating real-world data-to-insight agents.
> > > >
> > > > Following this feedback, we have implemented and begun evaluating two state-of-the-art multi-agent architectures:
> > > > * Reflexion (Shinn et al., 2023) — an iterative plan–critique–revise framework.
> > > > * PDT (Planner → Decomposer → Tool Executor) — a hierarchical, tool-augmented architecture aligned with recent programmatic agent systems such as DSPy (Gao et al., 2023) and the natural-language data-preparation system AutoPrep (Li et al., 2025), both of which decompose tasks into structured subtasks and execute them through tool-based Python modules.
> > > >
> > > > Reflexion and PDT are considered state-of-the-art because they embody the two most empirically validated advances in modern LLM agents: iterative self-correction and hierarchical, tool-grounded workflow execution. Reflexion (Shinn et al., 2023) showed across established benchmarks that adding a structured plan → act → critique → revise loop reliably improves long-horizon task performance, making it a widely adopted baseline. PDT-style architectures follow the strongest trend demonstrated in systems like DSPy (Gao et al., 2023), where agents achieve significantly higher accuracy and stability by separating global planning, explicit decomposition, and precise tool-based execution instead of relying on a single monolithic LLM. Together, these two designs represent the leading approaches for making agents more reliable, controllable, and scalable.
> > > >
> > > > Both architectures are implemented using smolagents, ensuring consistency with our existing setup. Preliminary Reflexion (full input mode) results are complete, and full PDT evaluation is underway. Early evidence suggests that while multi-agent coordination can improve robustness and intermediate verification, these systems still struggle with the complex, multi-step pipelines represented in KramaBench.
> > > >
> > > > In the revised paper, we will include full results across all six domains, along with a comparison table and discussion.
> > > >
> > > > |  | Archaeology | Astronomy | Biomedical | Environment | Legal  | Wildfire | **Overall** |
> > > > |---|---|---|---|----|---|---|---|
> > > > | smolagents Reflexion (claude-3-7-sonnet-latest) |50%| 33.33%| 44.44%| 80%| 66.67%| 47.62%| 57.69%|
> > > >
> > > > KramaBench is a living benchmark with a public leaderboard, and we welcome further suggestions for additional multi-agent baselines.
> > > > While the author feedback phase is still ongoing, we are committed and open to perform any additional experiment and welcome any further feedback or suggestion to strengthen our submission and improve our research.

---

### Author Response · Authors · 2025-12-03
**General Responses**

### Response Summary

Following up on the comments received, we updated a revised version of the paper where we marked all changes in blue. Where relevant, we provide high-level summaries and updates to our discussion in each review thread.

### Benchmarking Multi-agent Systems

As mentioned during this rebuttal phase, we experimented with two multi-agent baselines, PDT and Reflexion. We implemented these systems with Claude3.7 and GPTo3 backend. The full results for the benchmark are the following:

| System                 | Model       | Archaeology | Astronomy | Biomedical | Environment | Legal  | Wildfire | Overall |
|------------------------|-------------|-------------|-----------|------------|-------------|--------|----------|---------|
| smolagents-reflexion DR | Claude-3-7  | 41.67%      | 5.97%     | 42.32%     | 59.05%      | 59.27% | 60.26%   | 50.64%  |
| smolagents-reflexion DR | GPTo3       | 16.67%      | 13.44%    | 15.26%     | 3.04%       | 14.25% | 29.25%   | 14.69%  |
| smolagents-pdt DR       | Claude-3-7  | 25.00%      | 10.08%    | 2.22%      | 6.00%       | 10.22% | 36.98%   | 15.92%  |
| smolagents-pdt DR       | GPTo3       | 16.67%      | 2.46%     | 4.13%      | 0.68%       | 6.87%  | 26.50%   | 10.17%  |

We updated the text of the paper to include a discussion of these baselines and of their results. Due to space constraints, Appendix C of the paper provides the implementation and design details for the multi-agent baselines.

### Analysis of Performance Gaps
As mentioned above, the updated revision of the paper contains a more thorough discussion of the performance gaps highlighted by the reviewer. Due to space constraints, we report this analysis in Appendix D.

### Human Baseline
We completed a human data science study that assessed the human performances in solving Kramabench. We revised the text of the paper to include a discussion and results of these experiments.
We revised Table 4 of the main paper to include the following results:

|  | Archaeology | Astronomy | Biomedical | Environment | Legal  | Wildfire | **Overall** |
|---|---|---|---|----|---|---|---|
| Human experts | 66.67% | 54.55% | 100.00% | 81.91% | 74.19% | 58.00% | 71.07% |

Across all human-authored solutions, we performed a detailed analysis of the underlying failure causes – which is detailed in Appendix F of the paper. The most common error types are:

*Incorrect pipeline design (46%)*: The largest category. These errors occur when, for example, experts mis-specified a join, aggregation rule, grouping key, or filtering logic. This suggests that the most cognitively demanding part of real-world data science is pipeline design, rather than implementation.

*Lack of domain knowledge (24%)*: Many tasks contain implicit domain assumptions (e.g., definitions of “violation,” mapping categorical labels). Experts often produced internally consistent but mismatched interpretations. This shows that even humans struggle with domain-specific task semantics.

*Incorrect inputs (12%)*: Tasks often require gathering information across multiple similarly named or structurally similar files, and even humans sometimes use wrong inputs. These errors reflect the challenge of navigating multi-file datasets.

*Incorrect answer format (9%)*: Some errors are due to having the final outputs in the wrong representation (e.g., units, rounding, formatting), which did not match the one requested by the task.

*Library/version issues (9%)*: Minor inconsistencies (e.g., pandas handling) that changed intermediate results enough to fail strict correctness checking.

Overall, the human errors mirror the dominant failure modes of LLM agents. This confirms that KRAMABENCH captures genuinely difficult, real-world data-to-insight tasks where even trained data scientists struggle with pipeline reasoning, multi-file navigation, and implicit domain assumptions.

### Detailed Cost Analysis
As mentioned in the above response, we completed a cost analysis for all the baselines we benchmarked. The revised version of the paper contains the complete results regarding the cost/performance of all baselines considered. Due to space constraints, we include this analysis in an appendix.

### Human Agreement and LLM-Evaluation Calibration
The revised version of the paper now contains the human calibration and agreement analysis for the LLM-Evaluation scores. Due to space constraints, we provide this analysis in appendix G, along with a detailed discussion of the metrics chosen in the benchmark.

### Statistical Significance of Results
We are currently running all repetitions for the experiments as mentioned earlier in our rebuttal comment. We will update the results in a future version of the paper and on our leaderboard. All tables will include confidence bounds based on the standard deviation of the results across the three rounds.

---

### Meta-Review · Area_Chair_YbQP · 2025-12-20

**Summary:**

**Paper summary.** This paper introduces KRAMABENCH, a benchmark that tests end-to-end “data lake → pipeline design → execution → insight” capability. Instead of evaluating one isolated skill (SQL, code generation, or planning), tasks require systems to locate relevant files, clean/integrate data, run analysis, and produce a final answer. The benchmark contains 104 tasks spanning 1700 real-world files across 6 domains, and the paper uses a multi-level evaluation setup to separate pipeline-design failures from sub-task implementation failures.

**What happened in the discussion.** Reviewers’ main concerns were about trust and completeness: motivation/positioning, missing multi-agent baselines, unclear validity of LLM-based judging, and lack of cost/statistical reporting. The authors responded in depth and added major missing evidence in the forum: (1) results for two multi-agent baselines (PDT and Reflexion), (2) a human expert study and an overall human baseline (plus a breakdown of human failure modes), (3) calibration and agreement analysis for the LLM judge, and (4) cost analysis and plans to report confidence bounds from repeated runs. A reviewer explicitly commented that most doubts were resolved and that they would raise their score to 6.

**My assessment as AC.** With the discussion updates, I think this is a strong benchmark contribution: the dataset scope is meaningful, the evaluation framework helps diagnose failure modes, and the authors addressed the biggest “is this benchmark reliable?” questions with concrete additions (human baseline and judge calibration). The remaining gaps (more domains/languages/baselines) are natural future work rather than blocking issues.

**Decision.** Accept (poster). The paper fills a real gap and now has enough evidence for validity and usefulness.

**Reviewer Concerns:**

- **Reviewer 4eDt (rating 6, confidence 4)**: Requested multi-agent baselines, broader diversity (e.g., multilingual/multimodal), and more runtime/hardware reporting. Authors added multi-agent baselines and cost analysis; multilingual/multimodal remains future work. **Status:** mostly resolved.
- **Reviewer WbPU (rating 4, confidence 3)**: Wanted multi-agent evaluation and more analysis for domain-level performance gaps. Authors added multi-agent baselines and expanded analysis. **Status:** largely resolved.
- **Reviewer fJHC (rating 4, confidence 4)**: Raised motivation/positioning, figure quality, judge calibration, and cost/statistics. Authors provided detailed updates; reviewer stated they would raise score to 6. **Status:** resolved.
- **Reviewer 9FHL (rating 6, confidence 3)**: Provided limited actionable feedback (“see summary”), so there is little to respond to. **Status:** N/A.

**Reviewer Scores:**

- **4eDt (6,4)**: Likely unchanged.
- **WbPU (4,3)**: Likely increased after the added baselines and analysis (I would expect ~5–6).
- **fJHC (4,4)**: Explicitly raised to 6 in the forum.
- **9FHL (6,3)**: Likely unchanged.

---

### Decision · Program_Chairs · 2026-01-26

Accept (Poster)